

# Assessing storm surge hazard and impact of sea level rise in Lesser Antilles-Case study of Martinique

Yann Krien[1], Bernard Dudon[1], Jean Roger[1,2], Gaël Arnaud[1], Narcisse Zahibo[1]

[1]Laboratoire de Recherches en Géosciences, Université des Antilles, Guadeloupe, 97157, France
[2]G-Mer Etudes Marines, Guadeloupe, 97118, France

*Correspondence to*: Yann Krien (ykrien@gmail.com)

**Abstract.** In the Lesser Antilles, coastal inundations from hurricane-induced storm surges cause great threats to lives, properties, and ecosystems. Assessing current and future storm surge hazard with sufficient spatial resolution is of primary interest to help coastal planners and decision makers develop mitigation and adaptation measures. Here, we use wave-current numerical models and statistical methods to investigate worst case scenarios and 100-year surge levels for the case study of Martinique, under present climate or considering a potential sea-level rise. Results confirm that the wave setup plays a major role in Lesser Antilles, where the narrow island shelf impedes the piling-up of large amounts of wind-driven water on the shoreline during extreme events. The radiation stress gradients thus contribute significantly to the total surge, up to 100 % in some cases. The non-linear interactions of sea level rise with bathymetry and topography are generally found to be relatively small in Martinique, but can reach several tens of centimeters in low-lying areas where the inundation extent is strongly enhanced compared to present conditions. These findings further emphasize the importance of waves for developing operational storm surge warning systems in the Lesser Antilles, and encourage caution when using static methods to assess the impact of sea level rise on storm surge hazard.



## 1 - Introduction

Coastal urbanization and industrialization in storm surge prone areas pose great challenges for adaptation and mitigation. Human and economic losses due to water extremes have considerably increased over the last decades (WMO 2014), and are expected to continue to do so in many areas worldwide because of coastal population growth (Neumann et al 2015) and
climate change impacts (sea level rise, deterioration of protecting marine ecosystems, potential increase in the frequency of extreme events, etc). It is therefore necessary to better assess current and future storm surge hazard to help decision makers regulate land use in coastal areas and develop mitigation strategies.

The Lesser Antilles are the first islands on the path of hurricanes that originate off the west coasts of Africa and strengthen during their travel across the warm waters of the tropical Atlantic Ocean. They are therefore regularly exposed to extremely
severe winds and waves causing great human and economic losses. At the heart of the Lesser Antilles Archipelago lies Martinique, a French insular overseas region which shares similar characteristics with neighboring islands, such as a relatively narrow island shelf, fringing coral reefs , mangrove forests, numerous bays and contrasted slope morphologies. Although Martinique has been relatively spared over the last decades compared to other islands such as Dominica or Guadeloupe, it still largely suffered from massive destructions in coastal areas due to hurricanes passing nearby (Durand et
al 1997, Pagney and Leone 1999, Saffache 2000, Léone, 2007, Duvat 2015). A recent example is hurricane DEAN (category 2), which struck the island in 2007, causing severe damages, especially along the exposed east coast (Barras et al 2008).

About 15 years ago, the French national meteorological service delivered a preliminary map of 100-year surge heights in Martinique (Météo France, 2002). These early results were of great interest and have been used extensively by coastal planners since then (Grau and Roudil, 2013). At that time, however, wave induced setup was not taken into account, so that
water levels were expected to be underestimated in areas exposed to waves. In subsequent years, no attempt was made to improve storm surge hazard assessment in Martinique. These preliminary results are thus still largely used as a reference by decision makers. Recent works rather investigated the impacts of historical events (e.g. Barras et al 2008), or the ability of numerical models to reproduce extreme water levels and inland flooding (Lerma et al 2014).

Here we investigate in greater details storm surge hazard in Martinique and derive more accurate 100-year surge heights and
maximum surge levels, using state-of-the-art numerical models and the statistical-deterministic approach of Emanuel et al (2006). We also conduct preliminary tests to investigate the impact of sea level rise (SLR) in the following decades. The present paper is organized as followed : after a short presentation of the study area (section 2), we describe the methodology (section 3) and the numerical model (section 4). Results and conclusions are shown in sections 5 and 6 respectively.

## 2 - Study area

Located in the heart of the Lesser Antilles (Figure 1, left panel), Martinique is a french mountainous island of about 390 000 inhabitants, with a remarkable variety of coastal environments (mangroves, cliffs, sandy coves, coral reefs, highly urbanized, etc) and contrasted sea bottom morphologies. The Atlantic coast is characterized by barrier and fringing coral reefs, as well as a gently dipping dissipating shelf promoting relatively large storm surges (Figure 1, right panel), whereas most of the Caribbean beaches are reflective, with waves propagating onshore without significant attenuation, except in the Bay of Fort
de France.





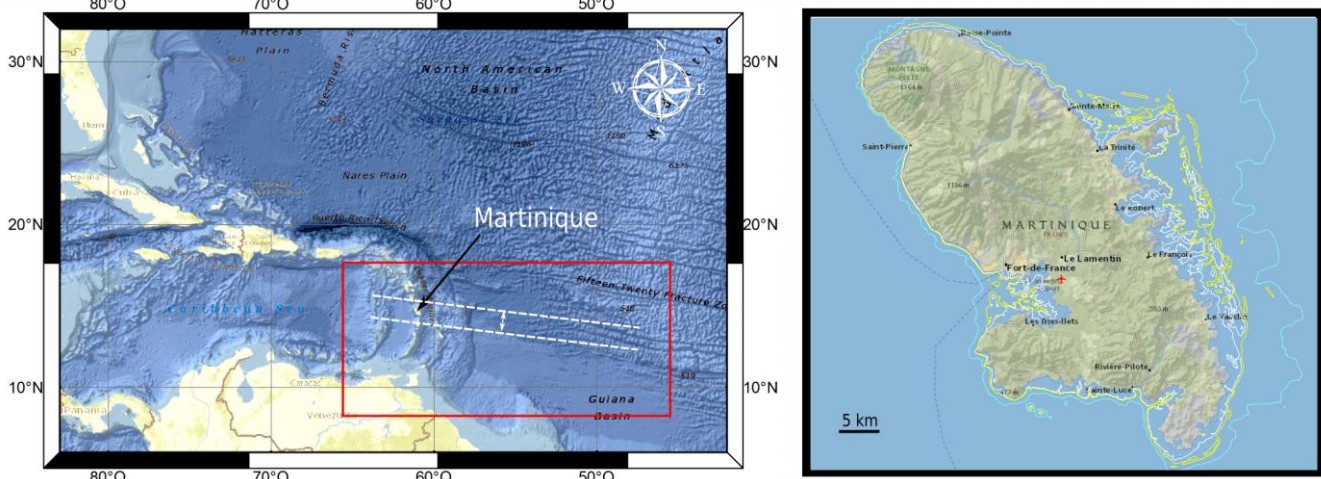

**Figure 1-Left panel : Area of interest. The computational domain is given in red. Dashed white lines represent the southernmost and northernmost tracks considered for the « worst case scenarios ». Right panel : focus on Martinique. The 10 m (white), 20 m (yellow), and 100 m (cyan) isobaths are displayed.**

A large part of the population has been living close to the shoreline for centuries, for historical or economical reasons, such as military defense or fishing activities (EPRI 2012). This trend is being accentuated with the development of tourism infrastructures since the 60's (Desarthe 2014). The number of tourists has thus tripled since 1995, even if the sector has undergone a deterioration recently (Dehoorne et al 2014). Coastal zones are now highly coveted and densely populated areas (Garnier et al 2015), prone to natural hazards such as erosion or storm surges. The Bay of Fort-de-France has been identified

as a particularly vulnerable area by the french government services, in the framework of the EU Floods Directive (PGRI 2014). Indeed, this relatively low-lying zone concentrate the great part of industry, services and transport infrastructures (highway, airport, etc). Besides, the mangrove forest of Lamentin is one of the largest remnant mangroves of the Caribbean and an important ecological area allowing the development of numerous animal species.

Martinique is regularly affected by severe storms, although it has been relatively spared over the past decades compared to
neighboring islands such as Dominica or Guadeloupe. The main event recorded in history is probably the hurricane that hit Martinique in 1780, resulting in about 9000 fatalities. More recently, the category 2 event DEAN (2007) caused very extensive damage to the urban areas close to the coast, as well as severe coastline erosion (Barras et al 2008). Significant destructions also arose in recent years because of energetic swells generated by hurricanes traveling eastward in the Caribbean Sea (e.g. OMAR in 2008 or Lenny 1999). The reflective Caribbean coast is particularly exposed to this type of
event. Hence, potential increase of hazards due to climate change (sea level rise, deterioration of protecting marine ecosystems, etc.) is a cause of major concern for the coming decades and strongly encourages to better assess current and future storm surge hazard along the coastline of Martinique.





## 3 - Methodology

To achieve this goal, we conducted numerical investigations using a wave-current coupled model (section 4). As a first step, we computed the maximum surge obtained for a few synthetic severe hurricanes. The aim is to better understand the mechanisms responsible for generating storm surges in Martinique, and to crudely estimate the maximum surges that could
be reached along the coastline for extreme events. To do this, we generated 13 synthetic hurricanes striking Martinique, with maximum velocity Vmax=140 kn, radius of maximum winds Rmax=20km, track angle of 10° with respect to a east-west profile, and translation speed Vt=12 kn. These values represent typical characteristics of major hurricanes in the area of interest (e.g. HUGO, DAVID ). The tracks parallel each other and are spaced 10 km apart to take into account almost all the possible landfall locations. The southernmost and northernmost tracks are displayed in Figure 1.

In a second phase, we derived new 100-year surge levels. This step is relatively complex for regions prone to cyclones because of the dearth of events in historical records. Traditional extreme value analysis methods are generally found not to be applicable in these areas, so that more advanced statistical approaches are needed to infer water level return periods. These methods involve the generation of a large number of synthetic cyclones that are in statistical agreement with
observations. Several approaches were proposed so far, such as JPM-OS (Joint Probability Methods with Optimal Sampling, e.g. Resio 2007, Toro et al 2010), or the statistical/deterministic model of Emanuel et al (2006). They have been used successfully for storm surge assessment at local (Lin et al 2010, 2012), regional (Harper et al 2009, Niedoroda et al 2010), or even continental (Haigh et al 2014) scales. In the present paper, we use the statististical-deterministic approach of Emanuel et al (2006), which provided good results for Guadeloupe in a previous study (Krien et al 2015). This method consists in four
main steps (Emanuel et al 2006) :

- 1-The genesis locations of the new synthetic storms are obtained by a random draw from a space-time probability density function derived from historical genesis point data.

- 2-For each storm considered, synthetic time series of the zonal and meridional wind components at 250hPa and 850hPa are generated. They are designed to conform to the climatologies derived from NCEP/NCAR reanalysis between 1980 and 2011. In particular, the observed monthly means and variances are respected, as well as most covariances. The wind time series are regenerated if the initial vertical shear is too strong to be conducive to a storm.
- 3-The storm track is then derived from a weighted mean of the 250- and 850-hPa flow, plus a correction for beta drift (Emanuel et al 2006). The weight factor and beta-drift term are chosen to optimize comparisons between the synthesized and observed displacement statistics.

- 4-The intensity along the synthetic track is obtained using a numerical model developed by Emanuel et al (2004). The wind shear is given by the synthetic time series of winds determined previously. Monthly mean climatological upper-ocean thermal structure is taken from Levitus (1982).

The full database used in this study contains 3200 low-pressure events (tropical depressions, tropical storms and
hurricanes) passing within 100 km from Fort de France. It represents about 8000 years of hurricane activity under the present climate conditions in the immediate vicinity of Martinique. In practice, however, we computed only the surges for the strongest events, as tropical storms and depressions are not found to able to generate water levels with a 100-yr return period. In all, 700 events were simulated on a 240-cores computational cluster.

In both cases, we also investigated the effect of a 1 m sea level rise. Considering that the sea level trend in the Lesser Antilles is very similar to the global mean rate (Palanisamy et al 2012), this value of 1 m corresponds to the global projections of IPCC by 2100 in case of a high emission scenario (IPCC, 2013). We assume here that coral reefs and mangroves will not been damaged but cannot keep pace with SLR. In practice, this amounts to rise the water level by 1 m, without changing the shape of bathymetry or topography.



## 4 - Numerical model

### 4.1-Model Description

In this study we employed the tightly coupled model ADCIRC+SWAN (Dietrich et al 2012). ADCIRC (ADvanced CIRCulation model, Luettich et al 1992, Westerink et al 1994) is a finite-element hydrodynamic model that solves the depth-
averaged barotropic form of the shallow water equations on unstructured grids. Water levels are obtained from the solution of the Generalized Wave-Continuity Equation (GWCE), whereas currents are derived from the vertically-integrated momentum equation. After several sensitivity tests, a timestep of 1 s was chosen.

ADCIRC is coupled to the wave model SWAN (Simulating WAves Nearshore, Booij et al 1999), which predicts the evolution in time and space of the wave action density spectrum, and has been converted recently to also run on unstructured
meshes (Zijlema et al 2010). Computations are performed here using 36 directions and 36 frequency bins. Source terms include wind input (Cavaleri and Malanotte-Rizzoli 1981, Komen et al 1984), quadruplet interactions (Hasselmann et al 1985), whitecapping (Komen et al 1984), triads (Eldeberky 1996), bottom friction (Madsen et al 1988) and wave breaking (Battjes and Janssen 1978).

SWAN is forced by the wind velocities, water levels and currents given by ADCIRC, and passes back the radiation stress
gradients every 10 minutes (Dietrich et al 2012). Bottom friction is computed in ADCIRC using a Manning formulation. Values depend on land cover (Union Europeenne, 2006), and can be found in Krien et al (2015). The coefficients are converted to roughness length by SWAN.

The model is forced by wind and pressure fields, calculated using the gradient wind profiles of Emanuel and Rotunno (2011) and Holland (1980) respectively (see Krien et al 2015 for more details).

Topography and bathymetry in shallow waters are specified using high-resolution LIDAR data (Litto3D Program). On the shelf, ship-based sounding data acquired by the French Naval Hydrographic and Oceanographic Department (SHOM) are also available. GEBCO (General Bathymetric Chart of the Oceans) data with 30-arc-second resolution are used for deep water areas.

The effect of tides are neglected here as their amplitude is very low in Martinique (less than 35 cm).

The computational domain is displayed in Figure 1. The resolution spans from 10 km in the deep ocean to about 50 m on the coastline and coral reefs.

### 4.2-Model performance

This model has been used and validated for various storm events around the world (e.g. Dietrich et al 2011a, 2011b, 2012; Hope et al., 2013, Kennedy et al., 2011, Murty et al. 2016). It was also found to give good results for several islands in the
Lesser Antilles, such as Guadeloupe and Martinique (Krien et al 2015, Lerma et al 2014).

In the course of the present study, we conducted a few more validation tests, such as for hurricane DEAN (2007). Results are consistent with observations, but those are not sufficiently accurate and compelling to add new elements to the storm surge validation process compared to previous studies. They are thus not displayed here (interested readers will find more
information in a report written in French : Krien 2013. As an example, the tide gauge located at Le Robert recorded a surge peak (of about 20 cm according to our estimates) on August 17, 2007, but this value is probably significantly underestimated since only hourly data are available. Similarly, only small surges (less than 20 cm) were recorded in Fort-de-France (Barras et al 2008) for hurricane DEAN. In the most impacted areas, such as Le Vauclin, only indirect informations about the maximum water level are available (e.g. Barras et al 2008). Hence, systematic measurements of water levels should be
performed in the future to be able to better validate and improve the model, as already stressed by Krien et al (2015).



## 5 - Results

### 5.1-Test cases for a few synthetic hurricanes and maximum surge levels

The results obtained for a few « worst case » tests are displayed in Figure 2. The water levels on the Caribbean coast are found to be largest for hurricanes making landfall in the northern part of Martinique. They can exceed 4 m above mean sea level in the upper part of the Bay of Fort-de-France for extreme events (Figure 2 (a)). In that case, most of the surge is driven by the wind. The wave setup only contributes for a few tens of centimeters to the total water levels (Figure 2(b)). This component plays yet a crucial role on the Atlantic coast, where it can reach 1 m. In some locations, such as Le Vauclin for example, the wave setup accounts here for almost all the total surge.

On the eastern coast, the surge is maximum for hurricanes passing south of Martinique. For category 4-5 hurricanes (such as the ones modelled here), it can exceed 3 m locally (Figure 2(c)). The wave setup is still significant (up to about 1 m) in the shallow waters between the coastline and the coral reefs on the Atlantic coast (Figure 2(d)). This contribution can amount to about 50 % of the total surge along the southeastern coasts of Martinique, in the test case considered here.

Figure 2(e) and Figure 2(f) show the results obtained when considering a sea level rise of 1 m. The wave setup is found to be only slightly modified, with a reduction of a few centimeters in general compared to the case without sea level rise Figure 2(d)). The wind-driven surge is significantly attenuated near the shore (by a few tens of centimeters), because wind stresses are less efficient in driving water masses towards the coast when the water depth is higher (comparison between Figure 2 (c) and Figure 2 (e)).



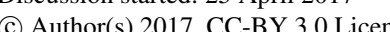

**Figure 2-Maximum water levels (left) and wave setup (right) for three « worst case » (category 4-5) hurricanes : northern track and no sea level rise ((a) and (b)), southern track and no sea level rise ((c) and (d)), and southern track with 1 m-sea level rise ((e) and (f)). The dashed black lines represent the track of the cyclone for each scenario. « Wave setup » refers here to the difference between the maximum water levels with and without waves.**





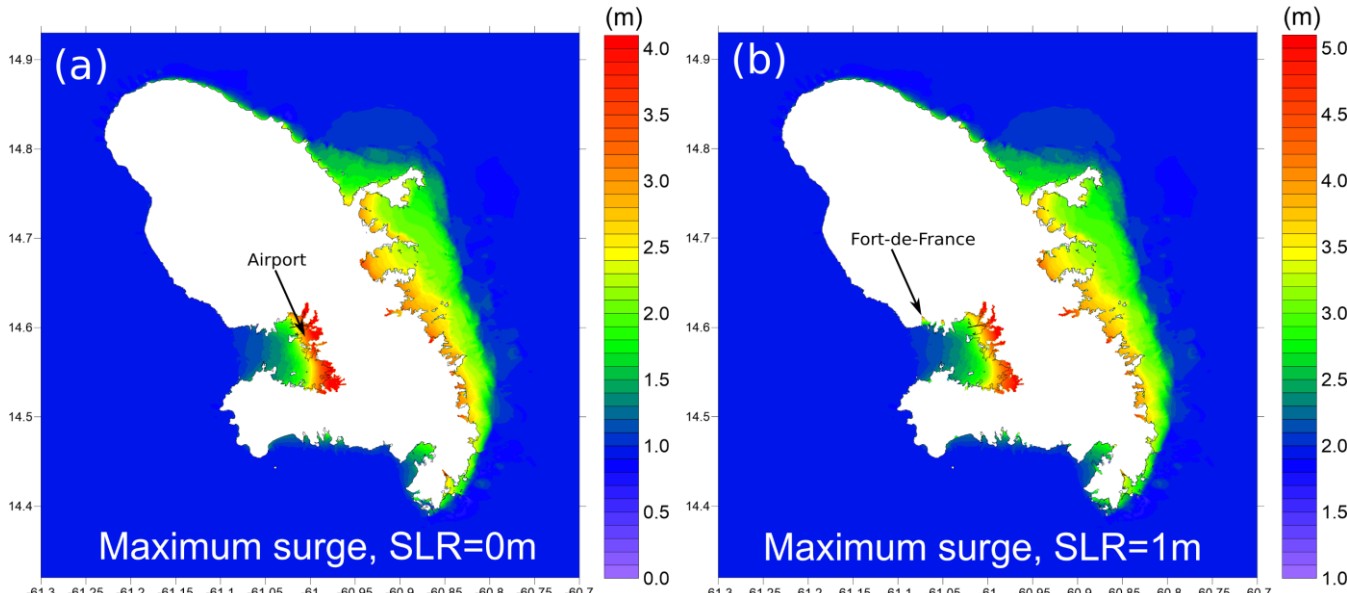

**Figure 3- Maximum surges obtained by considering « worst case » (category 4-5) hurricanes hitting Martinique, without (a) et with (b) sea level rise.**

The maximum water levels computed using the 13 synthetic category 4-5 hurricanes is presented in Figure 3(a). As mentioned above, the maximum surges are obtained for the bay of Fort-de-France, where water levels can exceed 4 m above mean sea level. The head of the bay, where high environmental and transportation stakes are located (e.g. airport, national roads, mangrove forest), is particularly exposed.

The shallow waters of the eastern coast also promote significant surges, which can reach about 3 m. On the other hand, the steep slopes characterizing most of the western part of Martinique impedes strong wind surges, and are generally not directly exposed to waves, so that maximum water levels are considerably reduced. Note however that the grid resolution (about 50m) is probably not sufficient to capture the wave setup in these areas, so that the maximum surge (about 1 m) might be somewhat underestimated.

A sea level rise of 1 m would have potentially major impacts, for example in the urban area of Fort-de-France (Figure 3(b)) where the cathedral or the courthouse could be flooded in case of a severe storm.

## 5.2- Statistical storm surge analysis

The 100-year surge levels obtained using the database described in section 3 are plotted in Figure 4(a). Since category 4-5 hurricanes striking Martinique (and for which maximum water levels can be reached locally) are rather scarce, they are found to be significantly smaller than the maximum surges estimated above, by a factor 2 or more. However, they still reach 1.5 m on the Atlantic coast, or in Fort-de-France's Bay. These 100-year surge heights are thus significantly higher than those computed in early studies (Météo France, 2002), with discrepancies that can reach 1 m for example in the south-east. Such differences are certainly largely due to the wave induced setup, which has been found to contribute significantly to the water levels (section 5.1), and was not accounted for in the early 2000's.

The impact of a 1 m sea level rise on 100-year surge levels is investigated in Figure 4(b). The non-linear interactions between surge and topography-bathymetry result in a decrease of water levels by several centimeters in most coastal areas, especially between the eastern shoreline and coral reefs, where the wave setup is reduced, and in shallow waters where the




wind is less efficient in generating surges because of larger water depths. Conversely, the 100-year surges are increased inland by a few tens of centimeters in low-lying regions where the inundation extent is strongly enhanced by the sea level rise (e.g. south-east of Fort-de-France Bay, or « Les Salines » beach, in the far south).

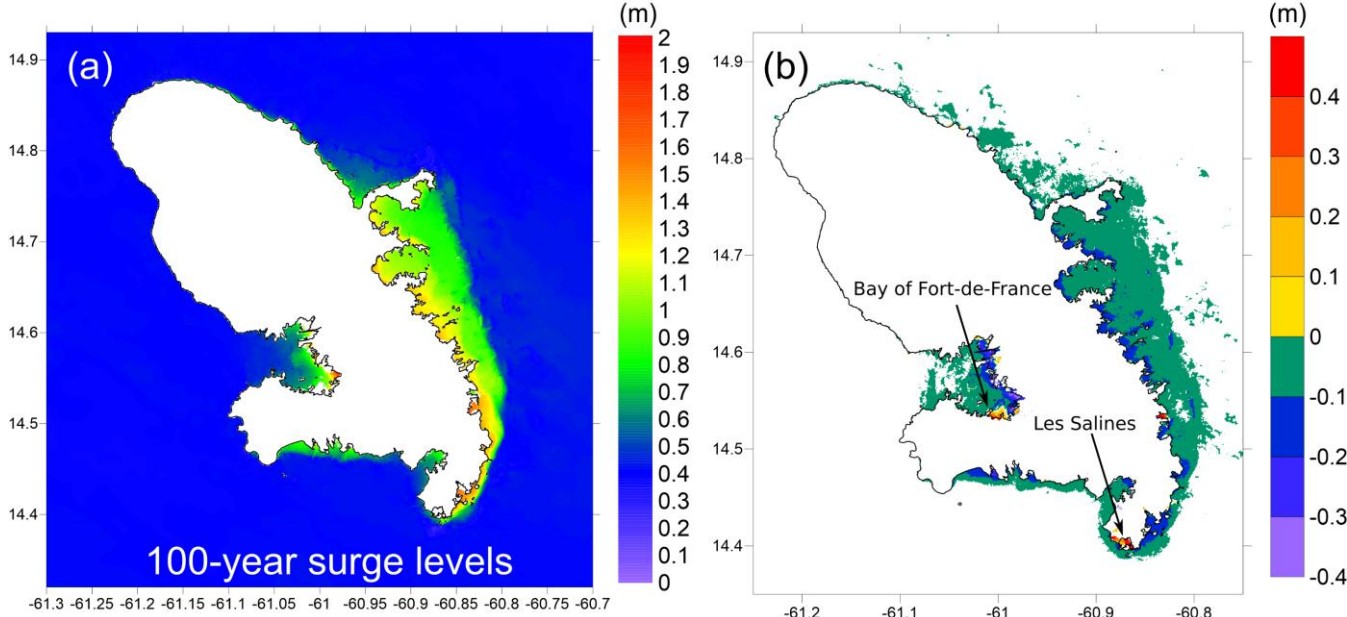

**Figure 4-100-year surge levels for present climate and no SLR (a), as well as difference between 100-year surge levels for present climate when considering a 1m-sea level rise (b).**

## 6 -Conclusions and discussion

Using coupled wave-current numerical models and a dataset of synthetic hurricanes representing thousands of years of cyclonic activity in the central part of the Lesser Antilles, we presented a detailed analysis of storm surge hazard in Martinique for the present climate, and started investigating the potential changes expected for the next decades. The 100-year and extreme surge levels are found to be highest for the bay of Fort-de-France and the Atlantic coast (south of La Trinité), where they can reach up to 4-5 m and 3 m respectively. In the latter case, a very significant part (up to about 1 m) of the surge is due to the wave setup. The contribution of the transfer of momentum from waves to the water column can amount up to 100 % of the total surge in some cases.

The non-linear interactions of sea level rise with bathymetry and topography are generally found to be relatively small, with a reduction of surge by a few centimeters in many nearshore areas, because the wave setup is reduced and the wind is less efficient in driving water masses towards the shoreline with increasing water depths. However, they can amount to several tens of centimeters in specific low-lying areas where the inundation extent is strongly enhanced compared to present conditions thanks to SLR. These results provide further evidence that drawing inundation maps with SLR by simply changing manually the elevation data (from a scenario without SLR) can lead to significant errors.

In case of a large sea level rise in the coming decades, hurricanes striking Martinique could have devastating impacts in the bay of Fort-de-France, where most economical, historical and transportation stakes are located. According to some of our «





worst case » scenarios, a large part of Fort-de-France urban area (including the cathedral or the courthouse) could be regularly flooded by hurricanes by the end of the 21st century. This finding also applies to the airport, and several major trunk roads.

These results should be of great help for policy makers and coastal planners to develop evacuation plans and implement adaptation measures. More work will be needed however to further investigate the impacts of climate change, including :

- *Changes in hurricane activity*. Although the effect of a warmer climate remains uncertain, a number of studies seem to reach the conclusion that the frequency of hurricanes will decrease, but that these events will be in average more intense (e.g. Knutson et al 2010). This might lead to changes in water levels for a given return period, even if preliminary works suggest that the impact could be very moderate compared to the effect of SLR (e.g. Condon and Sheng, 2012).

- *Evolution of coastal ecosystems*. Coral bleaching and mortality are expected to increase over the next decades due to ocean warming and acidification (e.g. Hoegh-Guldberg et al 2007, Baker et al 2008,Wong et al 2014). Besides, it is not clear whether coral reefs will be able to keep up with the sea level rise (we assumed in the present paper that it was not the case). This could have an impact in terms of surges, although the results presented here suggest that the effect might be moderate. It could also have major consequences in terms of wave impact at coastlines. Similarly, mangrove forests and seagrass beds could be sensitive to climate change (e.g. Waycott et al 2009, Gilman et al 2008), rendering shorelines more vulnerable to erosion and storm surges (e.g. Alongi, 2008, Wong et al 2014).

- *Evolution of the shoreline*, due to sediment transport, human activities or vertical motions.

These issues are currently being addressed for the French West Indies in the framework of C3AF, a project funded by the ERDF (European Regional Development Fund).

The methodology and results obtained here should be of interest for other islands in the Lesser Antilles, as they have similar morphological features as Martinique, such as a relatively narrow shelf, contrasted slope morphologies, presence of coral reefs and/or mangrove forests. This is confirmed for instance for the Guadeloupe archipelago, where very similar results in terms of 100-year surge levels (Krien et al 2015), maximum water levels, or wave setup contribution are found.

**Acknowledgments**

This work was supported by the INTERREG IV/TSUNAHOULE and FEDER/C3AF projects as well as Guadeloupe region. Many thanks to Kerry Emmanuel from Massachusetts Institute of Technology for providing the synthetic storm data sets, as well as to Raphaël Pasquier, Jacques Laminie and Pascal Poullet (University of the French West Indies) for the setup of the computing cluster.

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
