# Peer review of "Assessing storm surge hazard and impact of sea level rise in Lesser Antilles-Case study of Martinique"

_Natural Hazards and Earth System Sciences, 2017_

## Referee Comment (RC1) · Anonymous Referee #1 · 10 May 2017

Chapter 1 and 2: Dean as the most recent hurricane had strong impacts even though it was only category 2. Add a table that lists the main parameters of Dean and other recent (e.g. Hugo) or major historical hurricanes (e.g. 1780) that affected Martinique. Include parameters such as the distance of track from the shore, wind speeds and category and wave heights at closest position to the island, surge height, duration of inundation, inundation distance and height on land, flow speeds, . . . How many of the recent or historical hurricanes actually made landfall on Martinique? Are more storms passing north or south of the island? Add a map that depicts recent and historical hurricane tracks. https://coast.noaa.gov/hurricanes/ Setting: reefs and mangrove forests should be addressed in more detail, as they have a significant influence as

natural coastal protection. Reef type (fringing, patch)? Add the location of the reef and the mangroves in Fig. 1. In order to get an idea of the inundation risk: how wide are coastal plains on Martinique (max., min.)? A profile showing the bathymetry and onshore topography would be helpful, especially as the nearshore bathymetry has a significant influence on the potential storm surge and wave setup. Lines 17-18 (page 1): the importance of operational storm surge warning systems is stressed - are there plans for or already operating systems on Martinique? Have there been any evacuation trainings? How gets the population currently alerted? Are there any evacuations routes assigned? Line 13 (page 3): development is probably not the right term here - recolonization? Line 16 (page 3): add a reference for the 1780 event.

Chapter 3: Which hurricane categories were used? You mention cat 4-5 in chapter 5, but it should already be stated here. Line 39 (page 4): add a reference for the database; add a table with the input data used to set up your predictions. Line 46 (page 4): global mean sea level change (1 m) is cited which is very general. Are there any studies that predict a more local sea level? For a global or more local sea level: what are the max, average or min predictions?

Chapter 4: Is there any wave height data of Dean available? Which max heights and speeds were observed / measured? Line 32 (page 5): observations are mentioned but not explained – what was observed? Better add this to chapter 2. Why did the authors not select any additional scenario with an east-west track?

Chapter 5: Add a table that lists and compares the results of the 13 different scenarios. Include parameters like the modeled surge height, wave height, inundation distance, category, distance of storm track from the shore, and any other simulated parameters.

Chapter 6: Divide this chapter into separate chapters for the discussion and the conclusions. The discussion is far too short. For example it lacks a discussion on a Haiyan-like bore. The setting with a step bathymetry and reefs may allow the set up of infragravity waves on Martinique. Line 11 (page 10): the role of mangroves and reefs

as potential natural coastal protection has not been mentioned before. It should be discussed in greater detail. Also see comments to chapter 1 and 2.

Fig. 1 a and b have a bad resolution. White yellow and cyan lines can hardly be seen. Location of reefs and mangroves should be added.

---

## Referee Comment (RC2) · Anonymous Referee #2 · 15 May 2017

**General comments:**

The authors present an interesting study case paper about storm surges modeling at large scale. They deal with a complex and challenging topic which is the assessment of the storm surge hazard, including atmospheric and wave process in cyclonic context. I think this paper can be a substantial contribution but need some major reworking before publication. Generally, the bibliography is not enough developed. Second, a more in deep analysis of the results must be done and a discussion must be written in order to propose more convincing conclusions. Some figures need to be rework in order to be more explicit and serve the demonstration. Citation form must be harmonized in text and corrected according to the review recommendations.

Specific comments:

The introduction must be consolidated. Many studies using modelling tools and coupled models were achieved last years. The methods and tools used in this paper must be contextualize relative to the abundant bibliography on the topic. The paper deal with the influence of sea level rise on storm surge. Nevertheless there is no considerations made about climate change manifestations on coastal hazards especially on inundation risk. No information either on sea level rise trends in the Caribbean basin for example. Finally, a more in deep presentation of historical cyclones knowledge will be useful for the reader.

p.2/l.15 : Harmonize and correct citation form p.2/l.30 : Replace heart by center

Figure 1 : The figure need to be rework. Location are not readable, scale are not homogeneous. The figure don't look realized from raw data. Site the source of the maps if there are not fully realized by the authors.

p.3/l.9 : What about tsunami potential impact ?

p.3/l.16 : Precise that Dean were a category 5 hurricane but only category 2 when it circulate near to the Martinique island.

p.3/l.20 : The logical connection between the two sentences is not appropriated.

p.4/I.8-9 : Can you plot the historical track for comparison ? Land fall simulated tracks are not clearly visible on the figure.

p.4/l.39-43 : Unless I am misinterpreting, the full data base contain various cyclone intensity and trajectory. We suggest that it will be relevant to illustrate the track and the intensity of the cyclone contained in the data base.

p.5/I.3 : Tightly don't look as the proper term.

p.5/I.7 : You speak about sensitivity tests, explain more in detail what was the objective of the test and what elements are validated and what deduction are made.
p.5/I.20 : LIDAR data to what depth ?

p.5/I.22 : Are also available/used ?

p.5/l.25: Integration in Figure 1 of the mesh and the a representation of the special variability of the friction coefficient would be very interesting for interpretation of the results.

p.5/I.30 : Modify reference Lerma et al 2014 to Nicolae Lerma et al., 2014

p.5/I.35-38 : It is not clear whether the model reproduces correctly the observations made during the hurricane Dean (even if an only small storm surge was recorded). The model does represent this small storm surge or a more important values? In this section, the author need to be more explicit even if the validation is only qualitative.

Please explain more in detail the conclusion of the report Krien 2013.

p.6/I.7 : Locations mentioned in text must be placed on the plot (figure 2).

p.6/l.3-8 : Is not clear why the storm surge is so much higher in the Bay of Fortde-France for northern track than for the southern. Please, give a more complete explanation.

p.6/l.13 : The values of 1m was used arbitrarily or based on some references ?

Figure 2 please put the main location in the figure

Figure 2,b,d,f : There is some strange pattern in the northern coast. Are they artefact due to the computational mesh? Please explain this.

Figure 3 : Based on the figures, the model look to allow inland overflowing (i.e. there is no inland boundaries). If this true, it would be relevant to mention it. It would be also important to figure the coastline without surge on the plot.

p.8/l.13-14 : Please be more precise about this. Is this an historic reference ? Why this reference is relevant instead of indicate a % of flooded urban areas, for example?
p.9/l.15-17 : not clear, please reformulate.

p.9/I.23 : What do you mean by Âń simply changing manually the elevation data Âż ?

p.10/l.1-3 : This affirmation should be illustrated by a figure. A zoom on the mentioned area for example.

p.10/I.4-5 : Do you think the spatial resolution of your model (50 m at the coast) is in accordance with this perspective (i.e. evacuation plan, coastal urbanism...) ?

Discussion:

An extended chapter must be dedicated to discuss the results and the methodology. Did you test the impact of a West to East track hurricane? In reference to historical events, the authors must consider the impact of this kind of event? More generally, what can be the effect of a different track than considerate here? The affirmation of the integration of the wave setup or wave process must be tempered. A 50 m mesh resolution at the coast can be insufficient ton properly represents wave setup component (in steep beach or in coral reef area for example). Furthermore, spectral wave model do not deal with infragravity wave which can be important in reef coast. It is surprising to refer to very precise urban site in order to describe results in case of seal level rise scenario. What is the purpose? The model is it considerate as efficient to simulate floods?

NHESSD

---

## Author Response (AR1)

**Reviewer 1**

- *"Chapter 1 and 2: Dean as the most recent hurricane had strong impacts even though it was only category 2. Add a table that lists the main parameters of Dean and other recent (e.g. Hugo) or major historical hurricanes (e.g. 1780) that affected Martinique. Include parameters such as the distance of track from the shore, wind speeds and category and wave heights at closest position to the island, surge height, duration of inundation, inundation distance and height on land, flow speeds,... How many of the recent or historical hurricanes actually made landfall on Martinique? Are more storms passing north or south of the island? Add a map that depicts recent and historical hurricane tracks. https://coast.noaa.gov/hurricanes/"*

⇒ Following your advice, we included a figure displaying historical hurricane tracks. This should help the reader finding the answer to most of the questions you ask (distance of track from the shore, wind speeds/category, number of hurricanes making landfall on Martinique, frequency of storms north and south of the island, etc). As for the other information (surge height, duration of inundation, inundation distance, height on land, flow speeds), they are unfortunately largely unknown. The only few data available are either described directly in the text (e.g lines 230-240 for DEAN) or in other papers mentioned in the manuscript (e.g. Krien et al., 2015; Krien, 2013).

- *"Setting: reefs and mangrove forests should be addressed in more detail, as they have a significant influence as natural coastal protection. Reef type (fringing, patch)? Add the location of the reef and the mangroves in Fig. 1. In order to get an idea of the inundation risk: how wide are coastal plains on Martinique (max., min.)? A profile showing the bathymetry and onshore topography would be helpful, especially as the nearshore bathymetry has a significant influence on the potential storm surge and wave setup."*

⇒ Again, we followed your advice and: 1-added the location of reefs and mangroves in figure 1 , 2-displayed profiles showing bathymetric and topographic features in a few areas of interest (figure 1).

- *"Lines 17-18 (page 1): the importance of operational storm surge warning systems is stressed - are there plans for or already operating systems on Martinique? Have there been any evacuation trainings? How gets the population currently alerted? Are there any evacuations routes assigned?"*

⇒ As far as we know, there is no already operating system for the whole Martinique. As for evacuation routes or trainings, the situation seems to be quite different from one municipality to the other, and evolves quickly with time. It is indeed a very interesting subject, but giving a precise overview of this matter in Martinique would require a specific study, and is well beyond the scope of the present paper.

- *"Line 13 (page 3): development is probably not the right term here -recolonization?"*

⇒ The sentence was reworded (lines 97-98...)

- *"Line 16 (page 3): add a reference for the 1780 event."*

⇒ We added a reference (lines 103-104)

- *"Chapter 3: Which hurricane categories were used? You mention cat 4-5 in chapter 5, but it should already be stated here."*

⇒ We stated this in chapter 3 in the new version

- *"Line 39 (page 4): add a reference for the database"*

⇒ There seem to be a misunderstanding:  we did not use an already existing database, we developed one specifically for this study. We made this point clearer in the new version (line 169).

- *"add a table with the input data used to set up your predictions."*

⇒ Are you referring to input data relative to synthetic extreme events? If so, most of the information is given at the beginning of section 3, but for the sake of clarity, we also included a table in section 5.1 in the new version. On the other hand, if you are referring to the database and the atmospheric model, information can be found in Emanuel et al (2006) and Emanuel et al. (2004).

- *"Line 46 (page 4): global mean sea level change (1 m) is cited which is very general. Are there any studies that predict a more local sea level? For a global or more local sea level: what are the max, average or min predictions?"*

⇒ We already quoted the work of Palanisamy et al., 2012 in the first version of the paper. This works shows that the sea level trend in the Lesser Antilles is very similar to the global mean rate. We are not aware of any study giving predictions for 2100 at the local scale. In any case, uncertainties are large, from a few tens of centimeters to several meters in the worst cases. The value we chose (1m) is, as you say, very frequently considered by scientists or coastal planners when the trend is similar to the global mean sea level rise, as it is the case here. We tried to make this clearer in the new version.

- *"Chapter 4: Is there any wave height data of Dean available? Which max heights and speeds were observed / measured? Line 32 (page 5): observations are mentioned but not explained – what was observed? Better add this to chapter 2."*

⇒ Yes indeed, there are wave height data available for Dean. We give greater details on this matter in the new version, and (generally speaking) reformulated the whole paragraph to make things clearer to the reader.

- *"Why did the authors not select any additional scenario with an east-west track?"*

⇒ We performed several sensitivity tests on the track angle. Results show almost no sensitivity to this parameter. As a consequence, we did not consider a purely east-west track here. As for west-east tracks (maybe it was your point here?) we did not study specifically the impact of this kind of event for several reasons:

> 1- As far as we know, a west to east track hurricane passing nearby Martinique has never been observed (according to historical data). Events passing farther from Martinique were recorded (e.g Lenny, or Omar), but they induced very low surges in low-lying and vulnerable areas (such as bay of Fort-de-France). Hence, the probability that the extreme levels

presented here will be significantly exceeded in low-lying areas by this kind of event can be considered very low.

2-Most of the damages due to hurricanes such as Lenny or Omar where due to wave impacts (overtopping) at the shoreline along the north-western coast. The study of these processes are beyond the scope of this paper, which essentially concentrate on low-lying (surge prone) areas.

- *"Chapter 5: Add a table that lists and compares the results of the 13 different scenarios. Include parameters like the modeled surge height, wave height, inundation distance, category, distance of storm track from the shore, and any other simulated parameters."*

$\Rightarrow$ We added a table in chapter 5 that lists and compares the results of all the scenarios. Inundation distances are not given since the resolution is not good enough to represent this parameter with a satisfactory accuracy.

- *"Chapter 6: Divide this chapter into separate chapters for the discussion and the conclusions."*

$\Rightarrow$ We modified the paper accordingly

- *"The discussion is far too short. For example it lacks a discussion on a Haiyanlike bore. The setting with a step bathymetry and reefs may allow the set up of infragravity waves on Martinique."*

$\Rightarrow$ We added a paragraph on IG waves in the new manuscript. Generally speaking, the discussion as a whole has been consolidated.

- *Line 11 (page 10): the role of mangroves and reefs as potential natural coastal protection has not been mentioned before. It should be discussed in greater detail. Also see comments to chapter 1 and 2.*

$\Rightarrow$The role of mangroves and reefs is now mentioned in chapter 2. It is also discussed in greater details in the concluding remarks.

- *Fig. 1 a and b have a bad resolution. White yellow and cyan lines can hardly be seen. Location of reefs and mangroves should be added.*

$\Rightarrow$ Figure 1 was modified accordingly

**Reviewer 2**

- "*The introduction must be consolidated. Many studies using modelling tools and coupled models were achieved last years. The methods and tools used in this paper must be contextualize relative to the abundant bibliography on the topic. The paper deal with the*

*influence of sea level rise on storm surge. Nevertheless there is no considerations made about climate change manifestations on coastal hazards especially on inundation risk.*"

⇒ Following the advice of the reviewer, we expanded the introduction and quoted a number of papers relative to coupled numerical models and impacts of climate change on storm surges.

- "*No information either on sea level rise trends in the Caribbean basin*"

⇒ This information was given at the end of section 3 ("Considering that the sea level trend in the Lesser Antilles is very similar to the global mean rate (Palanisamy et al 2012)". But for the sake of clarity, we also added the reference (Palanisamy et al 2012) in the introduction and in section 2.

- "*A more in deep presentation of historical cyclones knowledge will be useful for the reader*"

⇒ We included a figure representing the track of historical cyclones since 1900 (figure 2). We also gave more details in the text.

- "*p.2/l.15 : Harmonize and correct citation form*"

⇒ We corrected the citation form, here and in other places in the manuscript

- "*p.2/l.30 : Replace heart by center*"

⇒ Corrected

- "*Figure 1 : The figure need to be rework. Location are not readable, scale are not homogeneous. The figure don't look realized from raw data. Site the source of the maps if there are not fully realized by the authors*"

⇒ Figure 1 has been modified and should be now more readable

- "*p.3/l.9 : What about tsunami potential impact ?*"

⇒ Indeed, Martinique is prone to tsunamis. We mentioned that and added a reference in the new version of the manuscript.

- "*p.3/l.16 : Precise that Dean were a category 5 hurricane but only category 2 when it circulate near to the Martinique island.*"

⇒ we clarified this in the new version

- "*p.3/l.20 : The logical connection between the two sentences is not appropriated.*"

⇒ Indeed, we corrected this in the new version

- "*p.4/l.8-9 : Can you plot the historical track for comparison ? Land fall simulated tracks are not clearly visible on the figure.*"

⇒ Historical events are too far from Martinique to be plotted on this Figure. However, we added a new figure displaying the historical tracks, so that it will be easier for readers to compare. The thickness of tracks was modified to make them more visible.

- *"p.4/l.39-43 : Unless I am misinterpreting, the full data base contain various cyclone intensity and trajectory. We suggest that it will be relevant to illustrate the track and the intensity of the cyclone contained in the data base."*

⇒we added a figure displaying a few examples of synthetic hurricanes contained in the database (figure 3)

- *"p.5/l.3 :Tightly don't look as the proper term."*

⇒ corrected

- *"p.5/l.7 : You speak about sensitivity tests, explain more in detail what was the objective of the test and what elements are validated and what deduction are made."*

⇒ we explained this in more details (lines 193-194)

- *"p.5/l.20 : LIDAR data to what depth ?"*

⇒ Up to about 40m depth. We specified this in the new version of the manuscript (line 212)

- *"p.5/l.22 : Are also available/used ?"*

⇒ Yes, they are available and included in the DEM. We specified this more clearly.

- *"p.5/l.25 : Integration in Figure 1 of the mesh and the a representation of the special variability of the friction coefficient would be very interesting for interpretation of the results."*

⇒The mesh is too highly resolved to be integrated in Figure 1 (the reader will not be able to distinguish between two different elements). Instead, we included a figure with a contour plot of the mesh resolution near Martinique, as well as a contour plot of the friction coefficient (figure 4).

- *"p.5/l.30 : Modify reference Lerma et al 2014 to Nicolae Lerma et al., 2014"*

⇒ corrected

- *"p.5/l.35-38 : It is not clear whether the model reproduces correctly the observations made during the hurricane Dean (even if an only small storm surge was recorded). The model does represent this small storm surge or a more important values? In this section, the author need to be more explicit even if the validation is only qualitative. Please explain more in detail the conclusion of the report Krien 2013."*

⇒ We significantly modified this section in order to be more explicit

- *"p.6/l.7 : Locations mentioned in text must be placed on the plot (figure 2)."*

⇒ We modified the figure accordingly

- *"p.6/l.3-8 : Is not clear why the storm surge is so much higher in the Bay of Fortde-France for northern track than for the southern. Please, give a more complete explanation."*

⇒ The reason is due to the direction of the wind when hurricanes pass over Martinique (offshore for southern tracks, onshore for northern track). We explained this more clearly in lines 248-249

- *"p.6/l.13 : The values of 1m was used arbitrarily or based on some references ?"*

⇒ We explain this in lines 178-180: " Considering that the sea level trend in the Lesser Antilles is very similar to the global mean rate (Palanisamy et al., 2012), this value of 1 m roughly corresponds to the global projections of IPCC by 2100 in case of a high emission scenario (IPCC, 2013)."

- *" Figure 2 please put the main location in the figure"*

⇒ We modified the figure accordingly

- *"Figure 2,b,d,f : There is some strange pattern in the northern coast. Are they artefact due to the computational mesh? Please explain this."*

⇒ Indeed, these patterns are probably due to small numerical instabilities in SWAN, in a region where lateral bathymetric gradients are strong. However, these errors are small since they do not exceed 1cm, so they are not expected to be an issue in our study. We added a comment on this in the figure legend.  NB: these numerical instabilities in unstructured SWAN were identified earlier by several authors. As far as we know, corrections have been made in the last versions of ADCIRC-SWAN to solve this issue.

- *"Figure 3 : Based on the figures, the model look to allow inland overflowing (i.e. there is no inland boundaries). If this true, it would be relevant to mention it.*

⇒ Indeed, the model includes a wetting-drying algorithm to allow inland overflowing. We mentioned this in the new version (line 192).

- *"It would be also important to figure the coastline without surge on the plot."*

⇒ the coastline is already represented by a thin black line. But we admit that it is not always easy to distinguish from the contour plots, so we increased its thickness in the figures.

- *"p.8/l.13-14 : Please be more precise about this. Is this an historic reference ? Why this reference is relevant instead of indicate a % of flooded urban areas, for example?"*

⇒ In Figure 3b, we wanted to show that there was a relatively significant area in Fort-de-France where a 1m sea level rise was expected to induce flooding. This was not clear enough apparently so we modified the figure to make it easier to understand.

- *"p.9/l.15-17 : not clear, please reformulate."*

⇒ We modified the text to make things clearer

- *"p.9/l.23 : What do you mean by ´n simply changing manually the elevation data Â˙z ?"*

⇒ We made this clearer in the new version of the paper: "These results provide further evidence that drawing inundation maps for the future without considering non-linear effects of sea level rise on water levels can lead to significant errors".

- *"p.10/l.1-3 : This affirmation should be illustrated by a figure. A zoom on the mentioned area for example."*

⇒ We illustrated this with a zoom in Figure 6(b).

- *p.10/l.4-5 : Do you think the spatial resolution of your model (50 m at the coast) is in accordance with this perspective (i.e. evacuation plan, coastal urbanism: : :) ?*

⇒ This resolution is definitely not sufficient to represent inland flooding properly. However, the water levels at the coastline should be relatively accurate in low-lying and shallow areas where surges are highest. This information is crucial for coastal urbanism or evacuation plans to identify the areas where vulnerable buildings should not be built, and to identify potential shelters for populations leaving close to the shoreline. To our knowledge, this work is so far the most elaborate study that can be used by coastal planners in France (even if we are fully aware of its limits!)

- *"Discussion: An extended chapter must be dedicated to discuss the results and the methodology. Did you test the impact of a West to East track hurricane? In reference to historical events, the authors must consider the impact of this kind of event?"*

⇒ A few west to east hurricanes are contained in the database, and are thus taken into account in the results presented in figure 7. However, we did not study specifically the impact of this kind of event for several reasons:

   1- As far as we know, a west to east track hurricane passing nearby Martinique has never been observed (according to historical data). Events passing farther from Martinique were recorded (e.g Lenny, or Omar), but they induced very low surges in low-lying and vulnerable areas (such as bay of Fort-de-France). Hence, the probability that the extreme levels presented here will be significantly exceeded in low-lying areas by this kind of event can be considered very low.

   2-Most of the damages due to hurricanes such as Lenny or Omar where due to wave impacts (overtopping) at the shoreline along the north-western coast. The study of these processes are beyond the scope of this paper, which essentially concentrate on low-lying (and surge prone) areas.

We discuss this matter in greater details in the new version of the manuscript.

- *"More generally, what can be the effect of a different track than considerate here?"*

⇒ According to Sansorgne (2013), a report written in French, the effect of track angle and translation speed for Martinique are of second order compared to hurricane intensity and distance to the area of interest (typically, a few percent of the total surge) . We mentioned this in the new version of the paper.

- *The affirmation of the integration of the wave setup or wave process must be tempered. A 50 m mesh resolution at the coast can be insufficient ton properly represents wave setup component(in steep beach or in coral reef area for example)*

$\Rightarrow$ Indeed, we already pointed this limitation in section 5.1 in the first version of the paper. This is true in particular in the north-western coast, where the slope is strong. In this case however, the stakes are more exposed to wave impacts (overtopping) than surges. As for the coral reefs, it is also true, although it seems to be less an issue, probably because: 1-the reefs are strongly eroded, so that bathymetric gradients are relatively mild 2-we ensured that the unstructured mesh captures the bathymetry in areas where the water depth is the lowest. We performed a few tests with better resolutions (e.g. 30m), without any significant changes. We included a section on this matter in the new version of the discussion.

- *Furthermore, spectral wave model do not deal with infragravity wave which can be important in reef coast.*

$\Rightarrow$ Again, you are right. Although large IG waves were not reported as such in Martinique, we see no reason why they should not occur. This has to be further investigated in the future. We included a section on this matter in the new version of the discussion.

- *It is surprising to refer to very precise urban site in order to describe results in case of seal level rise scenario. What is the purpose? The model is it considerate as efficient to simulate floods?*

$\Rightarrow$ We performed sensitivity tests with higher resolutions, up to 10m in some specific areas. In a quasi-systematic manner, flooded areas are larger in these cases. We also used phase-resolving models (SWASH) with extremely high resolutions (1m), with the same conclusion. Furthermore, the urban sites mentioned in the paper are very close to the shoreline, so even if the flooding dynamics is not perfectly well captured, our conclusions are expected to be correct. However, we understand your point, so that we modified the way to present our results in the new version of the manuscript.

**Assessing storm surge hazard and impact of sea level rise in Lesser Antilles-Case study of Martinique**

Krien[(1,*)], Y., Dudon[(1)], B., Roger[(1,2)], J., Arnaud[(1)], G., Zahibo[(1)], N.

*(1): LARGE, Laboratoire de Recherche en Géosciences, Université des Antilles, Guadeloupe, France*

*(2): G-Mer Etudes Marines, Guadeloupe, France*

*(*): Corresponding author (contact: ykrien@gmail.com)*

**Abstract**

In the Lesser Antilles, coastal inundations from hurricane-induced storm surges cause great threats to lives, properties, and ecosystems. Assessing current and future storm surge hazard with sufficient spatial resolution is of primary interest to help coastal planners and decision makers develop mitigation and adaptation measures. Here, we use wave-current numerical models and statistical methods to investigate worst case scenarios and 100-year surge levels for the case study of Martinique, under present climate or considering a potential sea-level rise. Results confirm that the wave setup plays a major role in Lesser Antilles, where the narrow island shelf impedes the piling-up of large amounts of wind-driven water on the shoreline during extreme events. The radiation stress gradients thus contribute significantly to the total surge, up to 100 % in some cases. The non-linear interactions of sea level rise with bathymetry and topography are generally found to be relatively small in Martinique, but can reach several tens of centimeters in low-lying areas where the inundation extent is strongly enhanced compared to present conditions. These findings further emphasize the importance of waves for developing operational storm surge warning systems in the Lesser Antilles, and encourage caution when using static methods to assess the impact of sea level rise on storm surge hazard.

**1-Introduction**

Coastal urbanization and industrialization in storm surge prone areas pose great challenges for adaptation and mitigation. Human and economic losses due to water extremes have considerably increased over the last decades (WMO, 2014), and are expected to continue to do so in many areas worldwide because of coastal population growth (Neumann et al., 2015) and climate change impacts (sea level rise, deterioration of protecting marine ecosystems, potential increase in the frequency of extreme events, etc). It is therefore necessary to better assess current and future storm surge hazard to help decision makers regulate land use in coastal areas and develop mitigation strategies.

The Lesser Antilles are the first islands on the path of hurricanes that originate off the west coasts of Africa and strengthen during their travel across the warm waters of the tropical Atlantic Ocean. They are therefore regularly exposed to extremely severe winds and waves causing great human and economic losses. In the center of the Lesser Antilles Archipelago lies Martinique, a French insular overseas region which shares similar characteristics with neighboring islands, such as a relatively narrow island shelf, fringing coral reefs, mangrove forests, numerous bays and contrasted slope morphologies.

Although Martinique has been relatively spared over the last decades compared to other islands such as Dominica or Guadeloupe, it still largely suffered from massive destructions in coastal areas due to hurricanes passing nearby (Durand et al., 1997; Pagney and Leone, 1999; Saffache, 2000; Léone,

2007; Duvat, 2015). A recent example is hurricane DEAN (category 2), which struck the island in 2007, causing severe damages, especially along the exposed east coast (Barras et al., 2008).

About 15 years ago, the French national meteorological service delivered a preliminary map of 100-year surge heights in Martinique (Météo France, 2002). These early results were of great interest and have been used extensively by coastal planners since then (Grau and Roudil, 2013). At that time, however, wave-current interactions were not taken into account, although waves were already known to have a strong impact on surges in coastal areas (e.g. Wolf et al., 2011; Brown et al., 2011). Water levels were thus expected to be underestimated, especially in areas exposed to waves.

Over the past few years, significant progress has been made in developing wave-current coupled models (e.g. Dietrich et al., 2012; Roland et al., 2012; Kumar et al., 2012; Qi et al., 2009; Bennis et al., 2011; Dutour Sikiric et al., 2013), but no attempt was made so far to improve storm surge hazard assessment in Martinique. These preliminary results are thus still largely used as a reference by decision makers, and recent works rather investigate the impacts of historical events (e.g. Barras et al., 2008), or the ability of numerical models to reproduce extreme water levels and inland flooding (Nicolae Lerma et al., 2014).

The potential impacts of climate change have also received little attention. Although the effect of a warmer climate remains relatively uncertain in terms of hurricane activity in the North Atlantic (e.g. Knutson et al., 2010), a significant increase of sea level is expected in the Lesser Antilles in the coming decades (Palanisamy et al., 2012). Moreover, coastal ecosystems such as mangroves, coral reefs or seagrass beds may not be able to adapt to climate change (e.g. Waycott et al., 2009; Wong et al., 2014), which could have large impacts on coastal flooding (e.g. Alongi, 2008; Wong et al., 2014).

Here we investigate in greater details storm surge hazard in Martinique and derive more accurate 100-year surge heights and maximum surge levels, using state-of-the-art numerical models and the statistical-deterministic approach of Emanuel et al (2006). We also conduct preliminary tests to investigate the impact of sea level rise (SLR) in the following decades. The present paper is organized as followed : after a short presentation of the study area (section 2), we describe the methodology (section 3) and the numerical model (section 4). Results and conclusions are shown in sections 5 and 6 respectively. The limitations of this study as well as material for further research are given in section 7.

2-Study area

Located in the center of the Lesser Antilles (Figure 1(a)), Martinique is a French mountainous island of about 390 000 inhabitants, with a remarkable variety of coastal environments (mangroves, cliffs, sandy coves, coral reefs, highly urbanized, etc) and contrasted sea bottom morphologies. The Atlantic coast is characterized by barrier and fringing coral reefs, as well as a gently dipping dissipating shelf promoting relatively large storm surges, whereas most of the Caribbean beaches are reflective, with waves propagating onshore without significant attenuation, except in the Bay of Fort de France (Figure 1).

[Figure]

Figure 1.-(a): Area of interest. The computational domain is given in red. Dashed white lines represent the southernmost and northernmost tracks considered for the « worst case scenarios » (section 5.1). (b): focus on Martinique and location of coral reef communities (red), mangroves areas (green), and lagoons (cyan). Source of data: Agence des aires marines protégées. (c): Isobaths at 10 m (white), 20 m (yellow), and 100 m (cyan), as well as location of the bathymetric profiles displayed in (d).

[Figure]

Figure 1 Left panel : Area of interest. The computational domain is given in red. Dashed white lines represent the southernmost and northernmost tracks considered for the « worst case scenarios ». Right panel : focus on Martinique. The 10 m (white), 20 m (yellow), and 100 m (cyan) isobaths are displayed.

A large part of the population has been living close to the shoreline for centuries, for historical or economical reasons, such as military defense or fishing activities (EPRI, 2012). This trend is being accentuated with the development of tourism infrastructures since the 60's (Desarthe, 2014). The number of tourists has thus tripled since 1995, even if the sector has undergone a deterioration recently (Dehoorne et al., 2014). Coastal zones are now highly coveted and densely populated areas (Garnier et al., 2015), prone to natural hazards such as erosion, or storm surges or tsunamis (e.g. Poisson and Pedreros, 2007). The Bay of Fort-de-France has been identified as a particularly vulnerable area by the frenchFrench government services, in the framework of the EU Floods Directive (PGRI, 2014). Indeed, this relatively low-lying zone concentrate the great part of industry, services and transport infrastructures (highway, airport, etc). Besides, the mangrove forest of Lamentin (Figure 1(b)) is one of the largest remnant mangroves of the Caribbean and an important ecological area allowing the development ofsupporting numerous a great variety of animal species.

Martinique is regularly affected by severe storms: sabout one hurricane every ten years on average, according to the data provided by NOAA's Office for Coastal Management (Figure 2) . Fortunately, i, although it has been relatively spared over the past decades compared to neighboring islands such as Dominica or Guadeloupe, with only one hurricane making landfall on the island since 1900 (Figure 2). The main event recorded in history is probably the hurricane that hit Martinique in 1780, resulting in about 9000 fatalities (Saffache et al., 2002). More recently, the category 2 event DEAN (-(2007)a category 5 hurricane that passed Martinique as a category 2 storm in 2007) caused very extensive damage to the urban areas close to the coast, as well as severe coastline erosion (Barras et al., 2008). Significant destructions also arose in recent years because of energetic swells generated by hurricanes traveling eastward in the Caribbean Sea (e.g. OMAR in 2008 or Lenny 1999). The reflective Caribbean coast is particularly exposed to this type of event.

[Figure]

**Figure 2. Tracks and intensities of historical hurricanes passing within 65 nautical miles from Martinique, since 1900. (source: NOAA's Office for Coastal Management, https://coast.noaa.gov/hurricanes/ )**

Marine ecosystems such as coral reefs or mangroves are known to provide substantial protection against waves and surges during hurricanes (e.g. Ferrario et al. 2014). In Martinique however, mangrove forests have been at least partially deteriorated due to earthworks, water and soil pollution, or hurricanes (e.g. Imbert and Migeot 2009). The situation is even worse for coral reefs, for which a dramatic decline due to eutrophication, anthropogenic disturbances or extreme storms was observed over the past 40 years (Bouchon and Laborel, 1986; Legrand et al., 2008; Rousseau et al., 2010;

IFRECOR, 2016). The deterioration of these marine ecosystems because of climate change is thus a cause Hence, potential increase of hazards due to climate change (sea level rise, deterioration of protecting marine ecosystems, etc.) is a cause of major concern for the coming decades.

A warmer climate is also expected to induce a significant increase of sea level in Martinique. Since the regional trends are very similar to the global mean rate (Palanisamy et al., 2012), the mean sea level might rise by several dozens of centimeters or more in the coming decades. All these findings and strongly encourages coastal planners and scientists to better assess current and future storm surge hazard along the coastline of Martinique in order to develop mitigation strategies.

3-Methodology

[revised manuscript text omitted]

4.2-Model performance

This model has been used and validated for various storm events around the world (e.g. Dietrich et al., 2011a, 2011b, 2012; Hope et al., 2013; Kennedy et al., 2011; Murty et al. 2016). It was also found to give good results for several islands in the Lesser Antilles, such as Guadeloupe and Martinique (Krien et al., 2015; Nicolae Lerma et al., 2014).

In the course of the present study, we conducted a few more validation tests, such as for hurricane DEAN (2007). Results are consistent with observations, but those are not sufficiently accurate and compelling to really add  relevant information regarding the ability of the model to reproduce storm surges. As an example, the tide gauge located at Le Robert recorded a surge peak (of about 20 cm according to our estimates) on August 17, 2007, but this value is probably significantly underestimated since only hourly data are available. Our model predicts higher values (about 75 cm), which are more consistent with observations made by witnesses, who reported that the garden south of city center (14.6753°N, 60.9387°W) was partially under water. Similarly, only small surges (less than 20 cm) were recorded in Fort-de-France (Barras et al., 2008) for hurricane DEAN. This is again consistent with the model prediction (15 cm), but not really satisfying in terms of validation for extreme events. Similarly, in the most impacted areas, such as Le Vauclin, only indirect information about the maximum water level are available (e.g. Barras et al., 2008). Although they are again in accordance with the predictions of the numerical model (about 1.5 m above mean sea level), systematic measurements of water levels should thus be performed in the future to be able to better assess the ability of the model to reproduce storm surges. Note that preliminary validation tests were also performed for waves, and give satisfying results (Krien 2013).

~~They are thus not displayed here (interested readers will find more information in a report written in French : Krien 2013). As an example, the tide gauge located at Le Robert recorded a surge peak (of about 20 cm according to our estimates) on August 17, 2007, but this value is probably significantly underestimated since only hourly data are available. Similarly, only small surges (less than 20 cm) were recorded in Fort-de-France (Barras et al 2008) for hurricane DEAN. In the most impacted areas, such as Le Vauclin, only indirect informations about the maximum water level are available (e.g.~~

Barras et al 2008). Hence, systematic measurements of water levels should be performed in the future to be able to better validate and improve the model, as already stressed by Krien et (2015).

5-Results

5.1-Test cases for a few synthetic hurricanes and maximum surge levels

The results obtained for a few « worst case » (category 4-5) eventstests are displayed in Figure 2Figure 5 and Table 1. The water levels on the Caribbean coast are found to be largest for hurricanes making landfall in the northern part of Martinique. This was expected since in this case, the winds on the west coast are essentially onshore when hurricanes pass over the island.

Table 1-Maximum storm surges (in meters) predicted by the model for the 13 "worst case" scenarios at four different locations: Fort-de-France tide gauge, airport, Le Robert tide gauge, and Le Vauclin. The distance of the hurricane track from Fort-de-France is also given. (S)/(N) refers to a storm passing south/north of Fort-de-France respectively. The maximum values obtained for each location are shown in bold.

| Test case | Distance from Fort-de-France (km) | Sea Level Rise (1m) | Surge Fort-de-France (61.063°W, 14.6°N) | Surge Airport (61.016°W, 14.593°N) | Surge Le Robert (60.937°W, 14.678°N) | Surge Le Vauclin (60.837°W, 14.548°N) |
|---|---|---|---|---|---|---|
| 1 | 59.7 (S) | no | 0.15 | 0.15 | 1.06 | 1.41 |
|   |          | yes | 0.14 | 0.13 | 0.96 | 1.36 |
| 2 | 48.7 (S) | no | 0.18 | 0.16 | 1.36 | 1.76 |
|   |          | yes | 0.16 | 0.15 | 1.24 | 1.68 |
| 3 | 37.8 (S) | no | 0.24 | 0.18 | 1.74 | 2.08 |
|   |          | yes | 0.21 | 0.17 | 1.61 | 2.04 |
| 4 | 26.9 (S) | no | 0.39 | 0.23 | 2.18 | 2.56 |
|   |          | yes | 0.36 | 0.21 | 2.04 | 2.44 |
| 5 | 16.0 (S) | no | 0.70 | 0.29 | 2.71 | **2.84** |
|   |          | yes | 0.69 | 0.29 | 2.55 | 2.71 |
| 6 | 5.1 (S)  | no | 1.09 | 0.85 | **3.04** | 2.54 |
|   |          | yes | 1.09 | 0.88 | 2.90 | 2.41 |
| 7 | 5.8 (N)  | no | 1.24 | 1.73 | 2.60 | 1.80 |
|   |          | yes | **1.26** | 1.67 | 2.49 | 1.68 |
| 8 | 16.7 (N) | no | 1.11 | **2.18** | 1.40 | 1.25 |
|   |          | yes | 1.09 | 1.99 | 1.43 | 1.18 |
| 9 | 27.7 (N) | no | 0.74 | 1.71 | 0.76 | 0.99 |
|   |          | yes | 0.75 | 1.59 | 0.77 | 0.95 |
| 10 | 38.6 (N) | no | 0.49 | 1.29 | 0.58 | 0.82 |
|    |          | yes | 0.50 | 1.17 | 0.60 | 0.77 |
| 11 | 49.5 (N) | no | 0.32 | 0.90 | 0.51 | 0.71 |
|    |          | yes | 0.32 | 0.79 | 0.51 | 0.69 |
| 12 | 60.4 (N) | no | 0.22 | 0.64 | 0.45 | 0.63 |
|    |          | yes | 0.21 | 0.54 | 0.45 | 0.62 |
| 13 | 71.3 (N) | no | 0.15 | 0.47 | 0.42 | 0.58 |
|    |          | yes | 0.14 | 0.38 | 0.41 | 0.56 |

. Water levelsThey can exceed 4 m above mean sea level in the upper part of the Bay of Fort-de-France for extreme events (Figure 2 (a)Figure 5 (a)). In that case, most of the surge is driven by the wind. The wave setup only contributes for a few tens of centimeters to the total water levels (Figure 2Figure 5(b)). This component plays yet a crucial role on the Atlantic coast, where it can reach 1 m. In

some locations, such as Le Vauclin for example, the wave setup accounts here for almost all the total surge.

On the eastern coast, the surge is maximum for hurricanes passing south of Martinique. For category 4-5 hurricanes (such as the ones modelled here), it can exceed 3 m locally (Figure 5(c)). The wave setup is still significant (up to about 1 m) in the shallow waters between the coastline and the coral reefs on the Atlantic coast (Figure 5(d)). This contribution can amount to about 50 % of the total surge along the southeastern coasts of Martinique, in the test case considered here.

Figure 5(e) and Figure 5(f) show the results obtained when considering a sea level rise of 1 m. The wave setup is found to be only slightly modified, with a reduction of a few centimeters in general compared to the case without sea level rise Figure 5(d)). The wind-driven surge is significantly attenuated near the shore (by a few tens of centimeters), because wind stresses are less efficient in driving water masses towards the coast when the water depth is higher (comparison between Figure 5(c) and Figure 5(e)).


[Figure]

Times New Roman

[Figure]

**Figure 5**-Maximum water levels (left) and wave setup (right) for three « worst case » (category 4-5) hurricanes : northern track and no sea level rise ((a) and (b)),  southern track and no sea level rise ((c) and (d)), and southern track  with 1 m-sea level rise ((e) and (f)). The dashed black lines represent the track of the cyclone for each scenario. « Wave setup » refers here to the difference between the maximum water levels with and without waves. Note that the wave setup "peaks" offshore the northwest coast are probably due to small numerical instabilities in SWAN, in a region with strong lateral bathymetric variations. Fortunately these errors are found to be very small (1 cm maximum) and bear no consequences on the results presented in this paper.

[revised manuscript text omitted]

Although tThese results constitute a significant step forward in assessing storm surge hazard and impacts of SLR in Martinique, should be of great help for policy makers and coastal planners to develop evacuation plans and implement adaptation measuresone must keep in mind that the study still leaves room for improvement. In particular,

mMore work will be needed however in the future to further investigate the impacts of climate change, including :

- *Changes in hurricane activity*.  Although the effect of a warmer climate remains uncertain, a number of studies seem to reach the conclusion that the frequency of hurricanes will decrease, but that these events will be in average more intense (e.g. Knutson et al., 2010). This might lead to changes in water levels for a given return period, even if preliminary works suggest that the impact could be very moderate compared to the effect of SLR (e.g. Condon and Sheng, 2012).
- *Evolution of coastal ecosystems*. Coral bleaching and mortality are expected to increase over the next decades due to ocean warming and acidification (e.g. Hoegh-Guldberg et al., 2007;, Baker et al., 2008;,Wong et al., 2014). Although it is not clear whether coral reefs will be able to keep up with the sea level rise in Martinique, their dramatic decline due to eutrophication, anthropogenic disturbances or hurricanes over the past 40 years (Bouchon and Laborel, 1986;

Legrand et al., 2008; Rousseau et al., 2010; IFRECOR, 2016) give little reason for optimism. This could have major consequences in terms of wave impacts at coastlines, and possibly also for surges, although the results presented here suggest that this effect might be moderate. Similarly, mangrove forests have been at least partially deteriorated due to earthworks, water and soil pollution, or hurricanes (e.g. Imbert and Migeot 2009), and may have difficulty adapting to climate change in some specific areas (Gilman et al., 2008; IFRECOR, 2016). Seagrass beds already degraded by anthropic pressure or patches of Sargassum (Thabard and Pouget-Cuvelier, 2014) might experience the same fate (Waycott et al., 2009). As a consequence, shorelines might be much more vulnerable to erosion and storm surges in the following decades (e.g. Alongi, 2008; Wong et al., 2014).

~~Besides, it is not clear whether coral reefs will be able to keep up with the sea level rise (we assumed in the present paper that it was not the case). This could have an impact in terms of surges, although the results presented here suggest that the effect might be moderate. It could also have major consequences in terms of wave impact at coastlines. Similarly, mangrove forests and seagrass beds could be sensitive to climate change (e.g. Waycott et al 2009, Gilman et al 2008), rendering shorelines more vulnerable to erosion and storm surges (e.g. Alongi, 2008, Wong et al 2014).~~

- *Evolution of the shoreline, due to sediment transport, human activities or vertical motions.* In Martinique, a few low sandy coastlines are subject to erosion and might be more exposed to relative sea level rise in the coming decades (Lemoigne et al 2013). This is the case for several coves, especially in the south (e.g. Sainte Anne, see Figure 1). However, most of low-lying coastal areas are rather in accretion, because of natural and/or anthropic factors. This has been observed in particular for the bay of Fort-de-France, where a coastline extension of about 100m was reported between 1951 and 2010 (Lemoigne et al 2013) in the mangrove area.
- Besides, the numerical approach can be further improved, particularly regarding:

- *The resolution.* Due to high computational costs, it was hardly possible to have a resolution better than 50m at the coastline and for coral reefs. To get an idea of the potential error on water levels, we performed a few sensitivity tests with higher resolutions (typically 20-30m). The discrepancy was found to amount only up to a few centimeters in shallow areas, where most of the stakes are exposed to storm surges. The coral reefs geometry seems to be satisfactorily captured by the mesh, probably because the reefs are strongly eroded (so that bathymetric gradients are relatively mild), and also because we ensured that the minimum water depths were correctly captured in these areas. However, a resolution of 50m is probably insufficient to properly assess the wave setup component in areas where the slope is steep. Results found are thus expected to be somewhat underestimated in the north-western coast for example. Note however that these areas are generally not really exposed to storm surges. There are more prone to wave overtopping, which is not taken into account in this study and will require further work in the future.
- *The phase-averaged model.* Phase-averaged models suffer from several limitations. In particular, they do not deal with run-up, which might contribute significantly to shoreline inundation (e.g Ford et al., 2013). This can be the case for example for fringing coral reefs, where the water level can be dominated by large low-frequency (e.g. infragravity) waves. Indeed, during extreme events, the spectral wave energy at reef crests shifts into lower frequencies, which can be amplified due to resonance modes (e.g. Roberts et al., 1992; Péquinet et al., 2009; Cheriton et al., 2016). Even if (to our knowledge) large infragravity

waves were not reported in Martinique, we see no reason to rule them out. Besides, climate change and sea level rise are expected to change the hydrodynamics across the reefs, and might further increase the exposure of coastlines to these type of waves (e.g. Merrifield et al., 2014). This issue has been receiving more and more attention over the last years, and will probably be a major topic of research in the near future.

- *West-to-East tracks.* A few hurricanes impacting Martinique and traveling eastward have been reported recently (e.g. OMAR in 2008 or Lenny in 1999). Although several synthetic events with similar characteristics are included in our database, these events might be to infrequent to be properly represented from a statistical point-of-view. Since they generally pass far away from Martinique, they are not expected to have large impacts on our computed 100-year surges in low-lying (surge prone) areas. But larger errors can be expected for steep slopes, on the western coast.

Some of these issues (impacts of climate change, resolution, etc) are currently being addressed for the French West Indies in the framework of C3AF, a project funded by the ERDF (European Regional Development Fund).

Note that tThe methodology and results obtained here should be of interest for other islands in the Lesser Antilles, as they have similar morphological features as Martinique, such as a relatively narrow shelf, contrasted slope morphologies, presence of coral reefs and/or mangrove forests. This is confirmed for instance for the Guadeloupe archipelago, where very similar results in terms of 100-year surge levels (Krien et al., 2015), maximum water levels, or wave setup contribution are found.

Acknowledgments

This work was supported by the INTERREG IV/TSUNAHOULE and FEDER/C3AF projects as well as Guadeloupe region. Many thanks to Kerry Emmanuel from Massachusetts Institute of Technology for providing the synthetic storm data sets, as well as to Raphaël Pasquier, Jacques Laminie and Pascal Poullet (University of the French West Indies) for the setup of the computing cluster.

| Page 2 : [1] Mis en forme | admin | 21/06/2017 14:12:00 |

Police :(Par défaut) Times New Roman, 11 pt

| Page 2 : [2] Mis en forme | admin | 21/06/2017 14:12:00 |

Police :(Par défaut) Times New Roman, 11 pt

| Page 2 : [3] Mis en forme | admin | 21/06/2017 14:12:00 |

Police :(Par défaut) Times New Roman, 11 pt

| Page 2 : [4] Mis en forme | admin | 21/06/2017 14:12:00 |

Police :11 pt, Anglais (États-Unis)

| Page 2 : [5] Mis en forme | admin | 21/06/2017 14:12:00 |

Police :(Par défaut) Times New Roman, 11 pt

| Page 2 : [6] Mis en forme | admin | 21/06/2017 14:12:00 |

Police :(Par défaut) Times New Roman, 11 pt

| Page 2 : [7] Mis en forme | admin | 21/06/2017 14:12:00 |

Police :(Par défaut) Times New Roman, 11 pt

| Page 2 : [8] Mis en forme | admin | 21/06/2017 14:12:00 |

Police :(Par défaut) Times New Roman, 11 pt

| Page 2 : [9] Mis en forme | admin | 21/06/2017 14:12:00 |

Police :(Par défaut) Times New Roman, 11 pt, Anglais (États-Unis), Non Surlignage

| Page 2 : [10] Mis en forme | admin | 21/06/2017 14:12:00 |

Police :(Par défaut) Times New Roman, 11 pt, Non Surlignage

| Page 2 : [11] Mis en forme | admin | 21/06/2017 14:12:00 |

Police :(Par défaut) Times New Roman, 11 pt, Anglais (États-Unis), Non Surlignage

| Page 2 : [12] Mis en forme | admin | 21/06/2017 14:12:00 |

Police :11 pt, Non Surlignage

| Page 2 : [13] Mis en forme | admin | 21/06/2017 14:12:00 |

Police :11 pt, Anglais (États-Unis)

| Page 2 : [14] Mis en forme | admin | 21/06/2017 14:12:00 |

Police :11 pt, Anglais (États-Unis)

| Page 2 : [15] Mis en forme | admin | 21/06/2017 14:12:00 |

Police :(Par défaut) Times New Roman, 11 pt

| Page 2 : [16] Mis en forme | admin | 21/06/2017 14:12:00 |

Police :(Par défaut) Times New Roman, 11 pt

| **Page 9 : [17] Mis en forme** | **admin** | **26/06/2017 12:00:00** |

Police :11 pt, Non Gras, Couleur de police : Automatique, Français (France)

| **Page 9 : [18] Mis en forme** | **admin** | **26/06/2017 10:12:00** |

Centré, Espace Après : 0 pt, Interligne : simple

| **Page 9 : [19] Mis en forme** | **admin** | **26/06/2017 10:12:00** |

Centré, Espace Après : 0 pt, Interligne : simple

| **Page 9 : [20] Mis en forme** | **admin** | **26/06/2017 10:12:00** |

Centré, Espace Après : 0 pt, Interligne : simple

| **Page 9 : [21] Mis en forme** | **admin** | **26/06/2017 10:12:00** |

Centré, Espace Après : 0 pt, Interligne : simple

| **Page 9 : [22] Mis en forme** | **admin** | **26/06/2017 10:12:00** |

Centré, Espace Après : 0 pt, Interligne : simple

| **Page 9 : [23] Mis en forme** | **admin** | **30/06/2017 17:59:00** |

Police :Gras, Soulignement

| **Page 9 : [24] Mis en forme** | **admin** | **26/06/2017 10:12:00** |

Centré, Espace Après : 0 pt, Interligne : simple

| **Page 9 : [25] Mis en forme** | **admin** | **30/06/2017 17:58:00** |

Police :Gras, Soulignement

| **Page 9 : [26] Mis en forme** | **admin** | **26/06/2017 10:12:00** |

Centré, Espace Après : 0 pt, Interligne : simple

| **Page 9 : [27] Mis en forme** | **admin** | **30/06/2017 17:58:00** |

Police :Gras, Soulignement

| **Page 9 : [28] Mis en forme** | **admin** | **26/06/2017 10:12:00** |

Centré, Espace Après : 0 pt, Interligne : simple

| **Page 9 : [29] Mis en forme** | **admin** | **30/06/2017 17:58:00** |

Police :Gras, Soulignement

| **Page 9 : [30] Mis en forme** | **admin** | **26/06/2017 10:12:00** |

Centré, Espace Après : 0 pt, Interligne : simple

| **Page 9 : [31] Mis en forme** | **admin** | **26/06/2017 10:12:00** |

Centré, Espace Après : 0 pt, Interligne : simple

| Page 9 : [32] Mis en forme | admin | 26/06/2017 10:12:00 |
|---|---|---|

Centré, Espace Après : 0 pt, Interligne : simple

| Page 9 : [33] Mis en forme | admin | 26/06/2017 10:12:00 |
|---|---|---|

Centré, Espace Après : 0 pt, Interligne : simple

| Page 9 : [34] Mis en forme | admin | 26/06/2017 10:12:00 |
|---|---|---|

Centré, Espace Après : 0 pt, Interligne : simple

---

## Author Response (AR2)

**Reviewer 1**

- *"The manuscript will be suitable for publication after the order of chapters 6 and 7 has been switched."*

⇒ Following the reviewer's advice, we switched chapters 6 and 7.

**Reviewer 2**

- *" Modifications made by the authors are satisfactory. Nevertheless, some technical point must be considerate for the final version. Figure 1 can still be improved in form. Scales are not systematically presented and in c) are presented twice. The figure appears a little bit untidy."*

⇒ We slightly modified Figure 1 to improve it. In particular, we added axis for b) and c), so that scales are now much clearer.

- *" Figure 6, the zoom at the upper right corner is too small to be analyzed. It deserves to be widened."*

⇒ The zoom was enlarged in Figure 6. However, we kept a relatively moderate size, because we do not want to give the impression that the resolution is high enough to give very accurate flooding patterns in urban areas, as already mentioned.

- *" Some citation are still presented in an incorrect form,"*

⇒ Indeed, we corrected a few typesetting errors. We hope we were able to remove all of them.

- *" The presentation form of the hurricane names are not homogeneous (e.g. LENNY or Lenny) ?"*

⇒ We corrected this in the new version of the manuscript.

**Assessing storm surge hazard and impact of sea level rise in Lesser Antilles-Case study of Martinique**

Krien[1,*], Y., Dudon[1], B., Roger[1,2], J., Arnaud[1], G., Zahibo[1], N.

*(1): LARGE, Laboratoire de Recherche en Géosciences, Université des Antilles, Guadeloupe, France*

*(2): G-Mer Etudes Marines, Guadeloupe, France*

*(*): Corresponding author (contact: ykrien@gmail.com)*

**Abstract**

In the Lesser Antilles, coastal inundations from hurricane-induced storm surges cause great threats to lives, properties, and ecosystems. Assessing current and future storm surge hazard with sufficient spatial resolution is of primary interest to help coastal planners and decision makers develop mitigation and adaptation measures. Here, we use wave-current numerical models and statistical methods to investigate worst case scenarios and 100-year surge levels for the case study of Martinique, under present climate or considering a potential sea-level rise. Results confirm that the wave setup plays a major role in Lesser Antilles, where the narrow island shelf impedes the piling-up of large amounts of wind-driven water on the shoreline during extreme events. The radiation stress gradients thus contribute significantly to the total surge, up to 100 % in some cases. The non-linear interactions of sea level rise with bathymetry and topography are generally found to be relatively small in Martinique, but can reach several tens of centimeters in low-lying areas where the inundation extent is strongly enhanced compared to present conditions. These findings further emphasize the importance of waves for developing operational storm surge warning systems in the Lesser Antilles, and encourage caution when using static methods to assess the impact of sea level rise on storm surge hazard.

**1-Introduction**

Coastal urbanization and industrialization in storm surge prone areas pose great challenges for adaptation and mitigation. Human and economic losses due to water extremes have considerably increased over the last decades (WMO, 2014), and are expected to continue to do so in many areas worldwide because of coastal population growth (Neumann et al., 2015) and climate change impacts (sea level rise, deterioration of protecting marine ecosystems, potential increase in the frequency of extreme events, etc). It is therefore necessary to better assess current and future storm surge hazard to help decision makers regulate land use in coastal areas and develop mitigation strategies.

The Lesser Antilles are the first islands on the path of hurricanes that originate off the west coasts of Africa and strengthen during their travel across the warm waters of the tropical Atlantic Ocean. They are therefore regularly exposed to extremely severe winds and waves causing great human and economic losses. In the center of the Lesser Antilles Archipelago lies Martinique, a French insular overseas region which shares similar characteristics with neighboring islands, such as a relatively narrow island shelf, fringing coral reefs, mangrove forests, numerous bays and contrasted slope morphologies.

Although Martinique has been relatively spared over the last decades compared to other islands such as Dominica or Guadeloupe, it still largely suffered from massive destructions in coastal areas due to

hurricanes passing nearby (Durand et al., 1997; Pagney and Leone, 1999; Saffache, 2000; Léone, 2007; Duvat, 2015). A recent example is hurricane Dean (category 2), which struck the island in 2007, causing severe damages, especially along the exposed east coast (Barras et al., 2008).

About 15 years ago, the French national meteorological service delivered a preliminary map of 100-year surge heights in Martinique (Météo France, 2002). These early results were of great interest and have been used extensively by coastal planners since then (Grau and Roudil, 2013). At that time, however, wave-current interactions were not taken into account, although waves were already known to have a strong impact on surges in coastal areas (e.g. Wolf et al., 2011; Brown et al., 2011). Water levels were thus expected to be underestimated, especially in areas exposed to waves.

Over the past few years, significant progress has been made in developing wave-current coupled models (e.g. Dietrich et al., 2012; Roland et al., 2012; Kumar et al., 2012; Qi et al., 2009; Bennis et al., 2011; Dutour Sikiric et al., 2013), but no attempt was made so far to improve storm surge hazard assessment in Martinique. The preliminary results are thus still largely used as a reference by decision makers, and recent works rather investigate the impacts of historical events (e.g. Barras et al., 2008), or the ability of numerical models to reproduce extreme water levels and inland flooding (Nicolae Lerma et al., 2014).

The potential impacts of climate change have also received little attention. Although the effect of a warmer climate remains relatively uncertain in terms of hurricane activity in the North Atlantic (e.g. Knutson et al., 2010), a significant increase of sea level is expected in the Lesser Antilles in the coming decades (Palanisamy et al., 2012). Moreover, coastal ecosystems such as mangroves, coral reefs or seagrass beds may not be able to adapt to climate change (e.g. Waycott et al., 2009;Wong et al., 2014), which could have large impacts on coastal flooding (e.g. Alongi, 2008; Wong et al., 2014).

Here we investigate in greater details storm surge hazard in Martinique and derive more accurate 100-year surge heights and maximum surge levels, using state-of-the-art numerical models and the statistical-deterministic approach of Emanuel et al (2006). We also conduct preliminary tests to investigate the impact of sea level rise (SLR) in the following decades. The present paper is organized as followed : after a short presentation of the study area (section 2), we describe the methodology (section 3) and the numerical model (section 4). The results are shown in section 5. The limitations of this study as well as material for further research are given in section 6. The main conclusions can be found in section 7.

2-Study area

Located in the center of the Lesser Antilles (Figure 1(a)), Martinique is a French mountainous island of about 390 000 inhabitants, with a remarkable variety of coastal environments (mangroves, cliffs, sandy coves, coral reefs, highly urbanized, etc) and contrasted sea bottom morphologies. The Atlantic coast is characterized by barrier and fringing coral reefs, as well as a gently dipping dissipating shelf promoting relatively large storm surges, whereas most of the Caribbean beaches are reflective, with waves propagating onshore without significant attenuation, except in the Bay of Fort de France (Figure 1).

[Figure]

[Figure]

Figure 1 -(a): Area of interest. The computational domain is given in red. Dashed white lines represent the southernmost and northernmost tracks considered for the « worst case scenarios » (section 5.1). (b): focus on Martinique and location of coral reef communities (red), mangroves areas (green), and lagoons (cyan). Source of data: Agence des aires marines protégées. (c): Isobaths at 10 m (white), 20 m (yellow), and 100 m (cyan), as well as location of the bathymetric profiles displayed in (d).

A large part of the population has been living close to the shoreline for centuries, for historical or economical reasons, such as military defense or fishing activities (EPRI, 2012). This trend is being accentuated with the development of tourism infrastructures since the 60's (Desarthe, 2014). The number of tourists has thus tripled since 1995, even if the sector has undergone a deterioration recently (Dehoorne et al., 2014). Coastal zones are now highly coveted and densely populated areas (Garnier et al., 2015), prone to natural hazards such as erosion, storm surges or tsunamis (e.g. Poisson and Pedreros, 2007). The Bay of Fort-de-France has been identified as a particularly vulnerable area by the French government services, in the framework of the EU Floods Directive (PGRI, 2014). Indeed, this relatively low-lying zone concentrate the great part of industry, services and transport infrastructures (highway, airport, etc). Besides, the mangrove forest of Lamentin (Figure 1(b)) is one of the largest remnant mangroves of the Caribbean and an important ecological area supporting a great variety of animal species.

Martinique is regularly affected by severe storms: about one hurricane every ten years on average, according to the data provided by NOAA's Office for Coastal Management (Figure 2) . Fortunately, it

has been relatively spared over the past decades compared to neighboring islands such as Dominica or Guadeloupe, with only one hurricane making landfall on the island since 1900 (Figure 2). The main event recorded in history is probably the hurricane that hit Martinique in 1780, resulting in about 9000 fatalities (Saffache et al., 2002). More recently, Dean (a category 5 hurricane that passed Martinique as a category 2 storm in 2007) caused very extensive damage to the urban areas close to the coast, as well as severe coastline erosion (Barras et al., 2008). Significant destructions also arose in recent years because of energetic swells generated by hurricanes traveling eastward in the Caribbean Sea (e.g. Omar in 2008 or Lenny 1999). The reflective Caribbean coast is particularly exposed to this type of event.

[Figure]

**Figure 2-Tracks and intensities of historical hurricanes passing within 65 nautical miles from Martinique, since 1900. (source: NOAA's Office for Coastal Management, https://coast.noaa.gov/hurricanes/ )**

Marine ecosystems such as coral reefs or mangroves are known to provide substantial protection against waves and surges during hurricanes (e.g. Ferrario et al., 2014). In Martinique however, mangrove forests have been at least partially deteriorated due to earthworks, water and soil pollution, or hurricanes (e.g. Imbert and Migeot, 2009). The situation is even worse for coral reefs, for which a dramatic decline due to eutrophication, anthropogenic disturbances or extreme storms was observed over the past 40 years (Bouchon and Laborel, 1986; Legrand et al., 2008; Rousseau et al., 2010; IFRECOR, 2016). The deterioration of these marine ecosystems because of climate change is thus a cause of major concern for the coming decades.

A warmer climate is also expected to induce a significant increase of sea level in Martinique. Since the regional trends are very similar to the global mean rate (Palanisamy et al., 2012),  the mean sea level might rise by several dozens of centimeters or more in the coming decades. All these findings strongly encourage coastal planners and scientists to better assess current and future storm surge hazard along the coastline of Martinique in order to develop mitigation strategies.

3-Methodology

To achieve this goal, we conducted numerical investigations using a wave-current coupled model (section 4). As a first step, we computed the maximum surge obtained for a few synthetic severe (category 4-5) hurricanes. The aim is to better understand the mechanisms responsible for generating storm surges in Martinique, and to crudely estimate the maximum surges that could be reached along

the coastline for extreme events. To do this, we generated 13 synthetic hurricanes striking Martinique, with maximum velocity Vmax=140 kn, radius of maximum winds Rmax=20km, track angle of 10° with respect to a east-west profile, and translation speed Vt=12 kn. These values represent typical characteristics of major hurricanes in Martinique (Sansorgne, 2013). A few sensitivity tests were performed to ensure that track angle and translation speed are of second order compared to hurricane intensity and distance to the area of interest (Sansorgne, 2013).

[revised manuscript text omitted]

After several sensitivity tests to achieve stability of model/results while keeping reasonable computing time, a timestep of 1 s was chosen.

ADCIRC (v50) is coupled to the wave model SWAN (Simulating WAves Nearshore, Booij et al., 1999), which predicts the evolution in time and space of the wave action density spectrum, and has been converted recently to also run on unstructured meshes (Zijlema et al., 2010). Computations are performed here using 36 directions and 36 frequency bins. Source terms include wind input (Cavaleri and Malanotte-Rizzoli, 1981; Komen et al., 1984), quadruplet interactions (Hasselmann et al., 1985), whitecapping (Komen et al., 1984), triads (Eldeberky, 1996), bottom friction (Madsen et al., 1988) and wave breaking (Battjes and Janssen, 1978).

SWAN is forced by the wind velocities, water levels and currents given by ADCIRC, and passes back the radiation stress gradients every 10 minutes (Dietrich et al., 2012). Bottom friction is computed in ADCIRC using a Manning formulation. The coefficients are converted to roughness length by SWAN. The Manning coefficient depends here on land cover (Union Europeenne, 2006). The values can be found in Krien et al. (2015), and are displayed in Figure 4(a). Note that we did not consider a strong dissipation of energy at the bottom for coral reefs, as they are known to be very eroded in Martinique (IFRECOR, 2016).

The model is forced by wind and pressure fields, calculated using the gradient wind profiles of Emanuel and Rotunno (2011) and Holland (1980) respectively (see Krien et al., 2015 for more details).

Topography and bathymetry in shallow waters (up to about 40 m depth) are specified using high-resolution Lidar data (Litto3D Program). On the shelf, ship-based sounding data acquired by the French Naval Hydrographic and Oceanographic Department (SHOM) are also included. GEBCO (General Bathymetric Chart of the Oceans) data with 30-arc-second resolution are used for deep water areas.

The effect of tides are neglected here as their amplitude is very low in Martinique (less than 35 cm).

The computational domain is displayed in Figure 1. The resolution spans from 10 km in the deep ocean to about 50 m on the coastline and coral reefs (Figure 4(b)).

[Figure]

Figure 4-(a): spatial variation of the Manning coefficient n, based on land cover data (Union Europeenne, 2006). (b): spatial variation of the mesh resolution in the vicinity of Martinique.

4.2-Model performance

This model has been used and validated for various storm events around the world (e.g. Dietrich et al., 2011a, 2011b, 2012; Hope et al., 2013; Kennedy et al., 2011; Murty et al., 2016). It was also found to give good results for several islands in the Lesser Antilles, such as Guadeloupe and Martinique (Krien et al., 2015; Nicolae Lerma et al., 2014).

In the course of the present study, we conducted a few more validation tests, such as for hurricane DeanEAN (2007). Results are consistent with observations, but those are not sufficiently accurate and compelling to really add relevant information regarding the ability of the model to reproduce storm surges. As an example, the tide gauge located at Le Robert recorded a surge peak (of about 20 cm according to our estimates) on August 17, 2007, but this value is probably significantly underestimated since only hourly data are available. Our model predicts higher values (about 75 cm), which are more consistent with observations made by witnesses, who reported that the garden south of city center (14.6753°N, 60.9387°W) was partially under water. Similarly, only small surges (less than 20 cm) were recorded in Fort-de-France (Barras et al., 2008) for hurricane DeanEAN. This is again consistent with the model prediction (15 cm), but not really satisfying in terms of validation for extreme events. Similarly, in the most impacted areas, such as Le Vauclin, only indirect information about the maximum water level are available (e.g. Barras et al., 2008). Although they are again in accordance with the predictions of the numerical model (about 1.5 m above mean sea level), systematic measurements of water levels should thus be performed in the future to be able to better assess the ability of the model to reproduce storm surges. Note that preliminary validation tests were also performed for waves, and give satisfying results (Krien 2013).

5-Results

5.1-Test cases for a few synthetic hurricanes and maximum surge levels

The results obtained for a few « worst case » (category 4-5) events are displayed in Figure 5 and Table 1. The water levels on the Caribbean coast are found to be largest for hurricanes making landfall in the northern part of Martinique. This was expected since in this case, the winds on the west coast are essentially onshore when hurricanes pass over the island.

Table 1-Maximum storm surges (in meters) predicted by the model for the 13 "worst case" scenarios at four different locations: Fort-de-France tide gauge, airport, Le Robert tide gauge, and Le Vauclin. The distance of the hurricane track from Fort-de-France is also given. (S)/(N) refers to a storm passing south/north of Fort-de-France respectively. The maximum values obtained for each location are shown in bold.

[revised manuscript text omitted]

**7 Discussion**

These results constitute a significant step forward in assessing storm surge hazard and impacts of SLR in Martinique. However, this study still leaves room for improvement. In particular, more work will be needed in the future to further investigate the impacts of climate change, including :

- *Changes in hurricane activity*. Although the effect of a warmer climate remains uncertain, a number of studies seem to reach the conclusion that the frequency of hurricanes will decrease, but that these events will be in average more intense (e.g. Knutson et al., 2010). This might lead to changes in water levels for a given return period, even if preliminary works suggest that the impact could be very moderate compared to the effect of SLR (e.g. Condon and Sheng, 2012).
- *Evolution of coastal ecosystems*. Coral bleaching and mortality are expected to increase over the next decades due to ocean warming and acidification (e.g. Hoegh-Guldberg et al., 2007; Baker et al., 2008;Wong et al., 2014). Although it is not clear whether coral reefs will be able to keep up with the sea level rise in Martinique, their dramatic decline due to eutrophication, anthropogenic disturbances or hurricanes over the past 40 years (Bouchon and Laborel, 1986; Legrand et al., 2008; Rousseau et al., 2010; IFRECOR, 2016) give little reason for optimism. This could have major consequences in terms of wave impacts at coastlines, and possibly also for surges, although the results presented here suggest that this effect might be moderate. Similarly, mangrove forests have been at least partially deteriorated due to earthworks, water and soil pollution, or hurricanes (e.g. Imbert and Migeot, 2009), and may have difficulty adapting to climate change in some specific areas (Gilman et al., 2008; IFRECOR, 2016). Seagrass beds already degraded by anthropic pressure or patches of Sargassum (Thabard and Pouget-Cuvelier, 2014) might experience the same fate (Waycott et al., 2009). As a consequence, shorelines might be much more vulnerable to erosion and storm surges in the following decades (e.g. Alongi, 2008; Wong et al., 2014).
- *Evolution of the shoreline, due to sediment transport, human activities or vertical motions*. In Martinique, a few low sandy coastlines are subject to erosion and might be more exposed to relative sea level rise in the coming decades (Lemoigne et al., 2013). This is the case for

several coves, especially in the south (e.g. Sainte Anne, see Figure 1). However, most of low-lying coastal areas are rather in accretion, because of natural and/or anthropic factors. This has been observed in particular for the bay of Fort-de-France, where a coastline extension of about 100m was reported between 1951 and 2010 (Lemoigne et al., 2013) in the mangrove area.

Besides, the numerical approach can be further improved, particularly regarding:

- *The resolution*. Due to high computational costs, it was hardly possible to have a resolution better than 50m at the coastline and for coral reefs. To get an idea of the potential error on water levels, we performed a few sensitivity tests with higher resolutions (typically 20-30m). The discrepancy was found to amount only up to a few centimeters in shallow areas, where most of the stakes are exposed to storm surges. The coral reefs geometry seems to be satisfactorily captured by the mesh, probably because the reefs are strongly eroded (so that bathymetric gradients are relatively mild), and also because we ensured that the minimum water depths were correctly captured in these areas. However, a resolution of 50m is probably insufficient to properly assess the wave setup component in areas where the slope is steep. Results found are thus expected to be somewhat underestimated in the north-western coast for example. Note however that these areas are generally not really exposed to storm surges. There are more prone to wave overtopping, which is not taken into account in this study and will require further work in the future.
- *The phase-averaged model*. Phase-averaged models suffer from several limitations. In particular, they do not deal with run-up, which might contribute significantly to shoreline inundation (e.g Ford et al., 2013). This can be the case for example for fringing coral reefs, where the water level can be dominated by large low-frequency (e.g. infragravity) waves. Indeed, during extreme events, the spectral wave energy at reef crests shifts into lower frequencies, which can be amplified due to resonance modes (e.g. Roberts et al., 1992; Péquinet et al., 2009; Cheriton et al., 2016). Even if (to our knowledge) large infragravity waves were not reported in Martinique, we see no reason to rule them out. Besides, climate change and sea level rise are expected to change the hydrodynamics across the reefs, and might further increase the exposure of coastlines to these type of waves (e.g. Merrifield et al., 2014). This issue has been receiving more and more attention over the last years, and will probably be a major topic of research in the near future.
- *West-to-East tracks.* A few hurricanes impacting Martinique and traveling eastward have been reported recently (e.g. Omar in 2008 or Lenny in 1999). Although several synthetic events with similar characteristics are included in our database, these events might be to infrequent to be properly represented from a statistical point-of-view. Since they generally pass far away from Martinique, they are not expected to have large impacts on our computed 100-year surges in low-lying (surge prone) areas. But larger errors can be expected for steep slopes, on the western coast.

~~Note that the methodology and results obtained here should be of interest for other islands in the Lesser Antilles, as they have similar morphological features as Martinique, such as a relatively narrow shelf, contrasted slope morphologies, presence of coral reefs and/or mangrove forests. This is confirmed for instance for the Guadeloupe archipelago, where very similar results in terms of 100-year surge levels (Krien et al., 2015), maximum water levels, or wave setup contribution are found.~~

**7 Conclusions**

Using coupled wave-current numerical models and a dataset of synthetic hurricanes representing thousands of years of cyclonic activity in the central part of the Lesser Antilles, we presented a detailed analysis of storm surge hazard in Martinique for the present climate, and started investigating the potential changes expected for the next decades. The 100-year and extreme surge levels are found to be highest for the bay of Fort-de-France and the Atlantic coast (south of La Trinité, see Figure 1 for location), where they can reach up to 4-5 m and 3 m respectively. A very significant part of the surge (up to about 1 m on the eastern coast) can be due to wave setup. The contribution of radiation stress gradients can even account for almost all the total surge in some special cases, for example for hurricanes making landfall in the northern part of Martinique that will induce essentially cross-shore or offshore winds (and hence low wind setup) on the south-eastern coast.

The non-linear interactions of sea level rise with bathymetry and topography are generally found to be relatively small, with a reduction of surge by a few centimeters in many nearshore areas, because the wave setup is reduced and the wind is less efficient in driving water masses towards the shoreline with increasing water depths. However, they can amount to several tens of centimeters in specific low-lying areas (mangroves or lagoons for example) where the inundation extent is strongly enhanced compared to present conditions, thanks to SLR. These results provide further evidence that drawing inundation maps for the future without considering non-linear effects of sea level rise on water levels can lead to significant errors.

In case of a large sea level rise in the coming decades, hurricanes striking Martinique could have devastating impacts in the bay of Fort-de-France, where most economical, historical and transportation stakes are located. According to some of our « worst case » scenarios, a large part of Fort-de-France urban area could be regularly flooded by hurricanes by the end of the 21st century. This finding also applies to the airport, located on the waterfront, and probably several major trunk roads.

The results presented in this paper can be further improved in terms of resolution, or by taking infragravity waves into account. The other impacts of climate change (evolution of coastal ecosystems and hurricane activity for example) could also be investigated in greater details. Some of those limitations are currently being addressed in the framework of C3AF, a project funded by the ERDF (European Regional Development Fund).

The methodology and results presented here should be of interest for other islands in the Lesser Antilles, as they display similar morphological features as Martinique, such as a relatively narrow shelf, contrasted slope morphologies, presence of coral reefs and/or mangrove forests. This is confirmed for instance for the Guadeloupe archipelago, where very similar results in terms of 100-year surge levels (Krien et al., 2015), maximum water levels, or wave setup contribution are found.

**Acknowledgments**

This work was supported by the INTERREG IV/TSUNAHOULE and FEDER/C3AF projects as well as Guadeloupe region. Many thanks to Kerry Emmanuel from Massachusetts Institute of Technology for providing the synthetic storm data sets, as well as to Raphaël Pasquier, Jacques Laminie and Pascal Poullet (University of the French West Indies) for the setup of the computing cluster. We also express our gratitude to Alexandre Nicolae Lerma and another anonymous reviewer for their helpful comments and suggestions, which have led to a significantly improved manuscript.